# Better Tokens for Better 3D: Advancing Vision-Language Modeling in 3D Medical Imaging

**Ibrahim Ethem Hamamci**[1,*]  **Sezgin Er**[1,*]  **Suprosanna Shit**[1,*]

**Hadrien Reynaud**[3]  **Dong Yang**[2]  **Pengfei Guo**[2]  **Marc Edgar**[2]  **Daguang Xu**[2]

**Bernhard Kainz**[3,4]  **Bjoern Menze**[1]

[1] University of Zurich  [2] NVIDIA  [3] Imperial College London  [4] FAU Erlangen-Nürnberg

## Abstract

Recent progress in vision-language modeling for 3D medical imaging has been fueled by large-scale computed tomography (CT) corpora with paired free-text reports, stronger architectures, and powerful pretrained models. This has enabled applications such as automated report generation and text-conditioned 3D image synthesis. Yet, current approaches struggle with high-resolution, long-sequence volumes: contrastive pretraining often yields vision encoders that are misaligned with clinical language, and slice-wise tokenization blurs fine anatomy, reducing diagnostic performance on downstream tasks. We introduce **BTB3D** (*Better Tokens for Better 3D*), a causal convolutional encoder-decoder that unifies 2D and 3D training and inference while producing compact, frequency-aware volumetric tokens. A three-stage training curriculum enables (i) local reconstruction, (ii) overlapping-window tiling, and (iii) long-context decoder refinement, during which the model learns from short slice excerpts yet generalizes to scans exceeding 300 slices without additional memory overhead. BTB3D sets a new state-of-the-art on two key tasks: it improves BLEU scores and increases clinical F1 by 40% over CT2Rep, CT-CHAT, and Merlin for report generation; and it reduces FID by 75% and halves FVD compared to GenerateCT and MedSyn for text-to-CT synthesis, producing anatomically consistent $512 \times 512 \times 241$ volumes. These results confirm that precise three-dimensional tokenization, rather than larger language backbones alone, is essential for scalable vision-language modeling in 3D medical imaging. The codebase is available at: `https://github.com/ibrahimethemhamamci/BTB3D`

## 1 Introduction

Three-dimensional (3D) medical images, such as CT, provide a rich volumetric view of anatomy, offering significantly more detail than 2D radiographs [1]. This makes them well-suited for vision-language models (VLMs), which can automate radiology report generation and enable text-guided volume synthesis for data augmentation, education, and planning [2]. Recent progress in 3D VLMs, driven by public datasets with paired reports (especially in chest CT) and advances in vision encoders and large language models (LLMs), has opened new clinical possibilities [3, 4, 5, 6, 7]. However, these systems still face major challenges: 3D volumes often consist of hundreds of slices, introducing long-sequence modeling challenges [8]. Moreover, the limited availability of paired 3D data prevents robust

---

*Authors may list their names first in their CVs. **Correspondence:** `ibrahim.hamamci@uzh.ch`

training [9]. As a result, 3D VLMs struggle to capture fine-grained clinical details and maintain spatial coherence when tasked with generating detailed reports or high-fidelity 3D volumes [10, 11, 12].

*Why do 3D VLMs lag behind their 2D counterparts in report generation?* A typical pipeline combines a pretrained 3D vision encoder with an LLM for report generation [3]. While LLMs have improved significantly through large-scale pretraining, the vision side remains a bottleneck [11]. Most state-of-the-art methods rely on vision encoders pretrained with contrastive objectives [13, 14]. However, this approach faces two critical limitations in the 3D medical context. First, contrastive learning assumes that only paired image-text samples are semantically aligned, while unpaired samples are unrelated [15]. In radiology, this assumption often does not hold: multiple reports can describe the same conditions differently due to variations in language, clinical focus, or radiologist style [16]. Penalizing such unmatched pairs may therefore degrade the model's understanding of medical semantics. Second, CLIP-style training requires large batch sizes for stability [17]. High-resolution 3D volumes are memory-intensive, often forcing trade-offs in batch size or model depth, resulting in underpowered vision encoders [14]. Prior work has noted that key findings (*e.g.*, small nodules or subtle textures) may be lost when compressing an entire 3D scan into a single vector using weak models trained contrastively [3]. Another major challenge is the lack of interoperability between 2D and 3D representations [3]. Most 2D medical VLMs cannot directly process volumetric input and instead require projecting 3D scans into 2D slices [18, 19]. This modality gap limits the transferability of pretrained 2D models to 3D, particularly important given the scarcity of 3D image-text datasets.

*A similar bottleneck affects text-conditional 3D medical image generation.* Prior methods rely on encoder-decoder networks pretrained in a self-supervised manner to reconstruct volumes [12]. However, training an encoder-decoder capable of reconstructing long, high-resolution CT sequences remains an open challenge. Thus, existing methods adopt cascaded generation frameworks, resulting in spatial discontinuities and reduced realism, or rely on lightweight encoder-decoder architectures that fail to capture nuanced context [20]. Consequently, current models struggle to generate anatomically consistent, high-quality 3D volumes from text. Hence, we need encoder-decoder networks for 3D VLMs that (1) scale to long volumetric sequences without losing critical details, (2) bridge the 2D/3D divide through unified representations, and (3) decode high-resolution 3D volumes from these representations. Improved encoders would enable more effective alignment between CT scans and textual descriptions for report generation, while better decoders enhance text-to-image generation.

To address these challenges, we introduce **BTB3D** (Better Tokens for Better 3D), a novel encoder-decoder framework that advances VLMs in 3D medical imaging through improved tokenization, reconstruction, and training strategies. Our core contribution is a *causal convolutional encoder-decoder* that learns a compact sequence of volumetric *tokens* for each CT. We adopt a quantized latent space for efficient 3D tokenization and reconstruction [21]. Causal 3D convolutions enable the model to process scans slice by slice (analogous to a temporal sequence), allowing scalability to arbitrarily long scans and compatibility with pretrained 2D features, serving as a *bridge between 2D and 3D modalities*. For downstream tasks, the proposed architecture can be combined with modules such as a transformer decoder for report generation or a diffusion model for volume synthesis. Crucially, we introduce a novel *three-stage training strategy* that progressively adapts the encoder-decoder to longer input contexts, enabling robust training even under compute-limited conditions.

## 2   Related works

**Radiology report generation from 3D medical images.**   The recent availability of 3D medical datasets paired with corresponding reports (such as CT-RATE) has enabled significant progress in this domain [3]. The first framework for radiology report generation from 3D scans, CT2Rep, did not leverage any pretrained vision or language models [10]. Due to limited training data and the need to generate long, meaningful reports, CT2Rep employed a relational memory module to retrieve relevant past reports for coherent generation, inspired by prior work [22]. Subsequent models, such as CT-CHAT and Merlin, improved upon this by incorporating pretrained components [3, 11]. These approaches followed a contrastive strategy to align image and text features during vision encoder pretraining [15]. A more recent model, fVLM, pretrained its 3D vision encoder, using a more clinically relevant contrastive learning approach [14]. However, we do not include fVLM in our comparisons, as neither its report generation codebase nor model weights have been available.

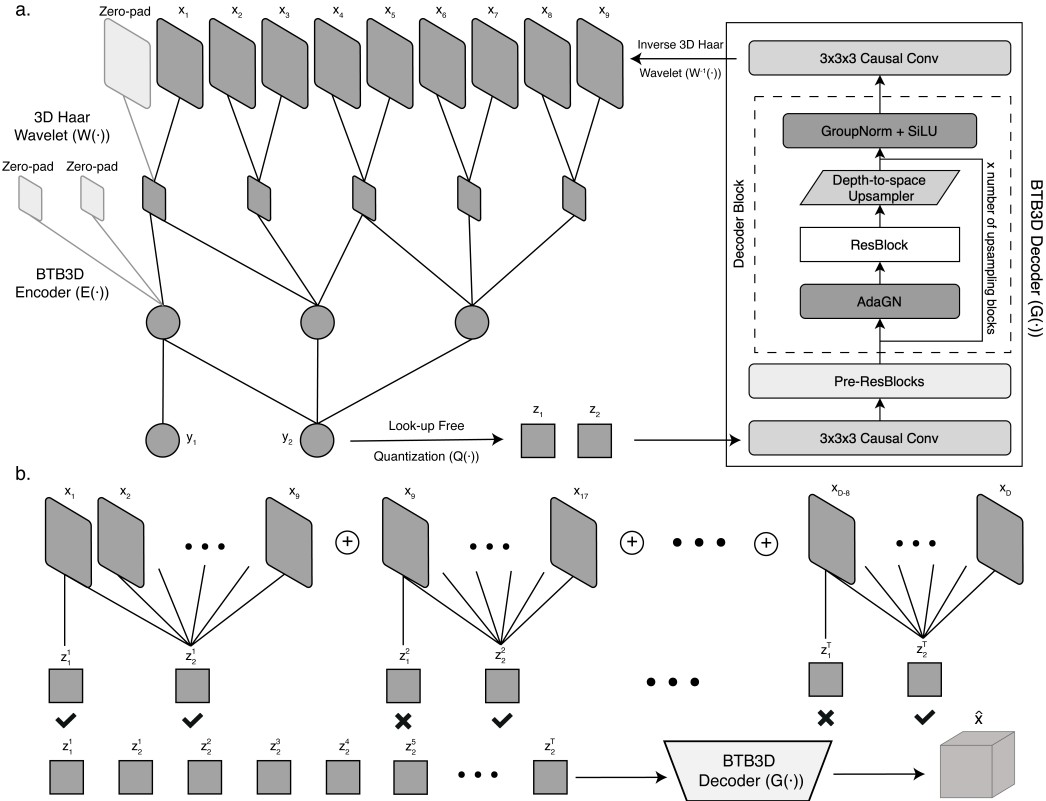

Figure 1: *(a)* Stage 1: A 9-slice subvolume is compressed via a wavelet transform, then encoded causally using two stride-2 temporal convolutions. The encoder processes the input strictly causally by prepending zero-padded slices. Tokens are decoded back to the wavelet domain by a symmetric causal decoder. *(b)* Stage 2: To scale to long CT volumes, we introduce overlapping temporal tiling (retaining only the second token from each window) to ensure consistent representation. Stage 3 follows the same scheme but trains only the decoder to refine long-range anatomical reconstruction.

**Text-conditional 3D medical image generation.** The first framework for text-conditioned 3D volume generation, GenerateCT, introduced a cascaded pipeline [12]. Due to computational limitations, it trained a 3D encoder-decoder network on low-resolution CT scans, followed by 2D diffusion steps for upsampling [23]. While this cascaded approach enabled high-resolution and long-sequence generation, the volumes suffered from poor interslice consistency. MedSyn addressed this limitation with a 3D architecture designed to improve spatial coherence; however, its lightweight design reduced image fidelity [20]. Other 3D CT scan generation methods, such as MAISI, exist, but we do not include them in our comparisons, as they are not designed for text-conditional generation [24].

## 3 Methodology

Due to the computational challenges, our goal is to pretrain on short sequences while enabling scalability to full volumes. Contrastive learning is inappropriate because paired reports typically describe the entire scan, and training on cropped or partial slices introduces semantic misalignment [13]. Also, they are difficult to adapt for 3D image synthesis [25]. Thus, we propose a reconstruction-based pretraining, generalizing effectively to longer sequences. Prior work has shown that convolutional networks outperform transformer-based ones in video reconstruction [26]. Thus, we adopt a 3D CNN-based backbone [27]. Given the limited 3D medical datasets, we design our model to support both 2D and 3D, enabling unified training across modalities, which motivates our causal network [28].

However, training a convolutional model on short sequences introduces two key challenges: (1) When performing inference in a single pass over the full CT, reconstruction quality tends to degrade toward the latter slices; or (2) When performing inference in short chunks, inconsistencies arise at window

boundaries. To address them, we introduce a three-stage training strategy that progressively scales while preserving temporal consistency. The resulting model can reconstruct long sequences with high fidelity and generalizes well across downstream tasks. Below, we detail the architecture and training.

## 3.1 Model architecture

BTB3D is a 3D convolutional encoder-decoder network with a discrete latent codebook [29]. We design it for CT scans using causal convolutional mechanisms along the temporal (axial) dimension and wavelet-based compression for efficient and scalable input representation, shown in Figure 1a.

**3D Haar wavelet compression.** Given a volumetric CT scan $x \in \mathbb{R}^{D \times H \times W}$, we apply a 3D Haar wavelet transform $W(\cdot)$ to obtain a multi-channel representation $W(x) \in \mathbb{R}^{\frac{D}{2} \times \frac{H}{2} \times \frac{W}{2} \times 8}$, following prior work in video generation that showed wavelet decompositions improve reconstruction quality [30, 31, 32]. This transformation reduces resolution by a factor of two along each axis while retaining essential frequency information. Each non-overlapping $2 \times 2 \times 2$ voxel block yields 8 subband coefficients: one low-frequency and seven high-frequency components. These frequency-aware channels encode both coarse and fine anatomical features, improving representation learning. Although the channel dimension increases from 1 to 8, the size of volumes is reduced by a factor of 8, yielding substantial compression for the input. This reduces memory and computation costs, enabling deeper or wider networks and making the model scalable to long, high-resolution CT volumes. For 2D slices ($D = 1$), the Haar transform along the $z$-axis produces a low- and high-frequency pair, with the latter often near-zero due to the absence of temporal variation. Nonetheless, the same 3D transform applies to both 2D and 3D inputs, ensuring architectural consistency across training modes.

**Encoder with causal 3D convolutions.** The encoder $E$ takes a wavelet-transformed CT sequence $W(x) \in \mathbb{R}^{D \times H \times W \times C}$ and outputs a latent representation $y \in \mathbb{R}^{D' \times H' \times W' \times d}$. It consists of residual blocks with *factorized 3D convolutions* that decouple spatial and temporal processing. Each block applies a $1 \times k \times k$ spatial convolution (sagittal-coronal) followed by a $k \times 1 \times 1$ temporal convolution (axial), with causal padding of $(k-1)$ zero slices in the past and none in the future. This ensures that the encoder at index $t$ only attends to slices $\leq t$, enabling strictly causal encoding. As a result, the first token is computed from the first slice alone, and future leakage is prevented in all subsequent tokens. This causal design supports unified 2D (e.g., single-slice) and 3D (volume-level) training while preserving axial consistency, beneficial for downstream autoregressive tasks [33, 34, 35].

Figure 1a illustrates the model architecture and causal axial compression. Downsampling is performed using strided convolutions interleaved with residual blocks. In the *8×8×8* configuration, two stride-2 convolutions along each axis yield a $4 \times$ reduction spatially and temporally. Combined with the initial $2 \times$ wavelet downsampling [31], this results in an effective $8 \times$ compression. The *16×16×8* variant adds an extra spatial stride-2 layer, achieving $16 \times$ spatial and $8 \times$ temporal compression. Each variant presents a trade-off: the $8^3$ setting preserves more spatial detail for tasks like segmentation or volume synthesis, while the $16^2 \times 8$ variant offers higher compression for memory-constrained settings or tasks focused on global semantics, such as classification or radiology report generation.

**Lookup-free quantization and decoder.** The encoder output $y = E(W(x)) \in \mathbb{R}^{D' \times H' \times W' \times d}$ is transformed into a discrete latent representation using *lookup-free quantization* [31]. The number of quantization codes is $K = 2^d$, and each feature vector at a spatial position is binarized independently across dimensions by computing the sign of each element, resulting in binary vectors $b \in \{-1, 1\}^d$. These vectors are then packed into integers, forming the token map $z \in \{0, \dots, K - 1\}^{D' \times H' \times W'}$. This approach removes the need for a codebook or embedding lookup during training and inference, significantly improving speed and memory efficiency, critical for large 3D volumes. To prevent code collapse (*e.g.*, overuse of a limited set of tokens), we include an entropy regularization term $\mathcal{L}_{\text{entropy}}$ that encourages uniform usage of all $K$ codes. Prior work in discrete representation learning [36, 26] has shown this regularization improves both reconstruction and token diversity. The decoder $G$, which mirrors the encoder's structure as shown in Figure 1a, employs transposed convolutions and residual upsampling blocks to reconstruct the wavelet-domain volume $\tilde{W}(x)$. Like the encoder, the decoder is also *causal*: each slice is reconstructed using only previously decoded slices, preserving autoregressivity [37]. The decoder accepts either the packed integer tokens or their binary expansions. The final volume $\hat{x}$ is obtained by applying $W^{-1}$, the inverse 3D Haar wavelet transform, to $\tilde{W}(x)$.

**Training objectives.** The model is trained to reconstruct high-resolution 3D CT sequences using a combination of three objectives: reconstruction loss, adversarial loss, and quantization loss.

The *reconstruction loss* $\mathcal{L}_{\text{rec}}$ encourages the decoder to accurately recover the input volume:

$$\mathcal{L}_{\text{rec}} = \mathbb{E}_x\left[\|x - \hat{x}\|_1\right] = \frac{1}{N}\sum_{i=1}^{N}|x_i - \hat{x}_i|, \tag{1}$$

where $\hat{x}$ is the reconstructed volume and $N$ is the number of voxels. We use the $\ell_1$ norm, which promotes sharper reconstructions and preserves fine anatomical detail better than $\ell_2$ [38].

The *adversarial loss* $\mathcal{L}_{\text{adv}}$ improves perceptual realism by encouraging indistinguishability:

$$\mathcal{L}_{\text{adv}} = \mathbb{E}_x\left[-\log D(\hat{x})\right] = -\frac{1}{N}\sum_{i=1}^{N}\log D(\hat{x}_i), \tag{2}$$

where $D$ is a 3D discriminator trained to distinguish real from generated CT volumes. We apply this supervision directly in the CT domain rather than the wavelet domain, stabilizing training [39, 36].

The *quantization loss* $\mathcal{L}_{\text{vq}}$ enforces commitment to discrete representations:

$$\mathcal{L}_{\text{vq}} = \mathbb{E}\left[\|\text{sg}[y] - e\|_2^2 + \beta\|y - \text{sg}[e]\|_2^2\right], \tag{3}$$

where $e$ is the discrete embedding (from a codebook or binary quantization), and $\text{sg}[\cdot]$ denotes the stop-gradient operator [40]. The overall training objective is a weighted sum:

$$\mathcal{L} = \mathcal{L}_{\text{rec}} + \lambda_{\text{adv}}\mathcal{L}_{\text{adv}} + \mathcal{L}_{\text{vq}}, \tag{4}$$

where $\lambda_{\text{adv}}$ controls the influence of adversarial supervision. We omit perceptual losses (*e.g.*, VGG-based features) due to the mismatch between natural RGB images and grayscale medical ones [41, 42]. Prior work has shown that these losses can degrade performance in medical image reconstruction [43].

## 3.2 Three-stage training

**Stage 1: Short-volume pretraining.** We first train the model on single 2D slices or short 9-slice subvolumes. The entire model (encoder, quantizer, and decoder) is optimized end-to-end:

$$\min_{E,G} \mathcal{L}(x_{1:9}, \hat{x}_{1:9}), \quad \text{where} \quad \hat{x}_{1:9} = W^{-1}(G(Q(E(W(x_{1:9}))))). \tag{5}$$

This phase helps the model learn local spatial and spatio-temporal structures.

**Stage 2: Overlapping temporal tiling.** We continue training both the encoder and decoder using overlapping short subsequences instead of full-volume encoding. To scale to long volumes while preserving temporal consistency, we adopt an overlapping window strategy. Specifically:

1. Encode $x_{1:9} \Rightarrow z_1^1, z_2^1$. Keep both tokens.
2. Encode $x_{9:17} \Rightarrow z_1^2, z_2^2$. Discard $z_1^2$ (covers only slice 9); keep $z_2^2$ (covers $x_9$–$x_{17}$).
3. Encode $x_{17:25} \Rightarrow z_1^3, z_2^3$. Discard $z_1^3$; keep $z_2^3$. Repeat this pattern until the end of the CT.

In each 9-slice window, we discard the first token and retain the second, which encodes information from all 9 slices (except the first window in which we take both tokens). This overlapping strategy promotes temporal consistency. Letting $z_i^t$ denote the $i$-th token from the $t$-th window, we have:

$$\text{1st window: } [z_1^1, z_2^1] = E(W(x_{1:9})) \tag{6}$$

$$\text{2nd window: } [z_1^2, z_2^2] = E(W(x_{9:17})) \Rightarrow \text{keep only } z_2^2 \tag{7}$$

$$\cdots$$

$$T\text{-th window: } [z_1^T, z_2^T] = E(W(x_{(D-8):D})) \Rightarrow \text{keep only } z_2^T \tag{8}$$

where $T = \lfloor (D-1)/8 \rfloor$, and $D$ is the sequence length of the partial CT volume that fits into memory in this second stage training. As shown in Figure 1b, the final latent sequence is:

$$[z_1^1, z_2^1, z_2^2, z_2^3, z_2^4, z_2^5, \ldots, z_2^T]. \tag{9}$$

This allows for efficient training on longer sequences than Stage 1, thanks to its lower memory footprint compared to one-shot encoding. The decoder processes all tokens in a single forward pass.

**Stage 3: Long-sequence decoder fine-tuning.** This stage mimics Stage 2, but we freeze the encoder $E$ and the codebook, and fine-tune only the decoder $G$. The training objective is:

$$\min_{\theta_G} \mathbb{E}_{x_{1:D}}\left[\mathcal{L}\left(x_{1:D}, \hat{x}_{1:D}\right)\right] \quad \text{subject to } E \text{ and codebook frozen,} \tag{10}$$

where $\theta_G$ are the parameters of the decoder $G$. This step enhances the decoder's capacity to model long-range anatomical dependencies by training it to reconstruct CT without modifying the encoder.

**Inference.** BTB3D supports two inference strategies for reconstructing full-length 3D CT volumes: *one-shot* and *tiled* inference, offering flexibility based on memory constraints and desired consistency. In *one-shot inference*, the entire volume $x$ is encoded and decoded in a single pass:

$$\hat{x} = W^{-1}(G(E(W(x)))). \tag{11}$$

This is efficient when memory allows full-volume processing but it may degrade on long sequences. In *tiled inference*, we follow the overlapping tokenization scheme from Stage 2 to ensure spatial coherence over long sequences. The input volume $x$ is split into overlapping 9-slice windows, and:

$$[z_1^t, z_2^t] = E(W(x_{s_t:s_t+8})), \quad \hat{x} = W^{-1}(G([z_1^1, z_2^1, z_2^2, z_2^3, \ldots, z_2^T])). \tag{12}$$

This strategy mirrors the training and ensures consistent reconstructions with lower memory use. We adopt tiled inference for all downstream experiments due to its robustness and improved consistency.

## 4 Experiments

We evaluate the effectiveness of our model and three-stage training through reconstruction and two downstream tasks (report generation and text-to-CT synthesis). Our experiments address three key questions: (1) Does the three-stage training improve reconstruction over naive end-to-end training? (2) Can our tokenization enhance vision-language tasks like report generation? (3) How does BTB3D compare to state-of-the-art models in synthesizing high-resolution CT from clinical prompts?

### 4.1 Three-stage training performance

We assess BTB3D's three-stage training via reconstruction metrics and ablations, demonstrating its scalability from short subvolumes to full-resolution 3D chest CTs. The strategy progressively (1) learns local spatial and short-range temporal features, (2) extends to longer sequences using overlapping tiling, and (3) enhances global coherence through decoder-only fine-tuning. This enables strong representations for downstream tasks such as report and text-to-CT generation (Sections 4.2 and 4.3).

Table 1: Reconstruction metrics after each stage of the three-stage training. We report full-volume reconstruction performance at each stage for two models with compression rates of 8×8×8 and 16×16×8.

| Stage | C. Rate | PSNR ↑ | SSIM ↑ | MSE ↓ |
|-------|---------|--------|--------|-------|
| Stage 1 | $8^3$ | 9.350 | 0.206 | 0.117 |
| Stage 2 | $8^3$ | 23.980 | 0.697 | 0.005 |
| Stage 3 | $8^3$ | **28.166** | **0.760** | **0.001** |
| Stage 1 | $16^2{\times}8$ | 11.067 | 0.353 | 0.079 |
| Stage 2 | $16^2{\times}8$ | 23.808 | 0.700 | 0.005 |
| Stage 3 | $16^2{\times}8$ | **26.750** | **0.749** | **0.002** |

**Experimental results.** Table 1 reports reconstruction performance across training stages [44, 45]. Stage 1, trained on short subvolumes, captures local structure but fails at long-range consistency. Stage 2 introduces overlapping tiling, yielding the largest improvement: PSNR increases by over 14 dB, SSIM triples, and MSE drops an order of magnitude. Stage 3 refines inter-slice fidelity through decoder tuning, with modest but consistent gains. Figure 2 visualizes reconstruction quality (8×8×8 variant) across planes. Stage 1 outputs are blurry and lose structure beyond 9 slices. Stage 2 restores coherent anatomy across distant slices. Stage 3 sharpens fine boundaries such as fissures and vessels.

**Dataset and implementation.** Aligned with the baselines, we use 25,692 chest CT scans from 21,304 patients in CT-RATE (the largest public 3D medical dataset with paired reports) [3]. The training set includes 20,000 patients and the test set includes 1,304. Volumes are converted to Hounsfield Units (HU) and clipped to $[-1000, 1000]$ [46, 47]. For Stages 1 and 2, we use raw volumes; for Stage 3, volumes are resampled to $0.75 \times 0.75 \times 1.5$ mm and cropped/padded to $512 \times 512 \times 241$. Training is conducted on 64 NVIDIA H100 GPUs using DDP and mixed precision.

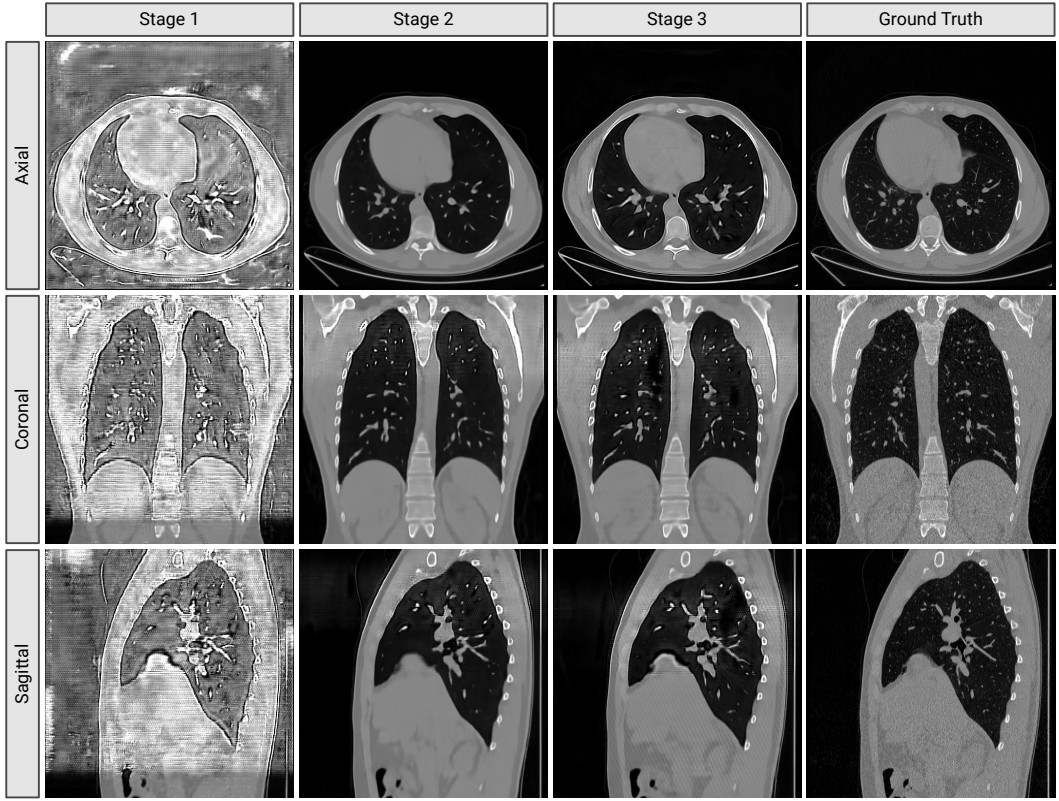

Figure 2: Reconstruction results of our $8{\times}8{\times}8$ model across axial, coronal, and sagittal planes after each training stage. Progressive improvements demonstrate the effectiveness of our three-stage strategy, with Stage 2 providing the largest gains in anatomical fidelity and inter-slice consistency.

In Stage 1, the batch size is 8 for 9-slice subvolumes and 40 for single slices. In Stages 2 and 3, we use a batch size of 1 (with 201 and 241 slices, respectively). The discriminator is reinitialized at each stage. We use Adam [48] with a learning rate of $1\mathrm{e}{-5}$, $\beta_2 = 0.99$, and $\epsilon = 1\mathrm{e}{-8}$, along with gradient clipping (threshold: 0.5) [49]. LFQ uses a token dimension of 18 (codebook size: 262,144) [31]. We train Stage 1 for 150k iterations, Stage 2 for 60k, and Stage 3 for 50k.

## 4.2 Radiology report generation from 3D chest CT

We evaluate BTB3D on report generation for 3D CT. Each volume is encoded into a sequence of latent tokens $\mathbf{v} \in \mathbb{R}^{T \times d_v}$ by the BTB3D encoder. For the $16 \times 16 \times 8$ model, we use 18-dimensional tokens; for the $8 \times 8 \times 8$ one, the token count is quadrupled and the embeddings are reduced to 72 via merging. A linear layer maps these tokens to the input space of LLaMA 3.1-8B [50], which autoregressively generates reports following the CT-CHAT setup [3]. We adopt the same configuration and pretrained LLM as CT-CHAT to ensure fair comparison and isolate the impact of our tokenization.

Table 2: Evaluation on Rad-ChestCT proves strong out-of-distribution generalization. As only binary labels are available, text-based metrics are omitted.

| Model | F1 ↑ | Precison ↑ | Recall ↑ |
|---|---|---|---|
| CT2Rep | 0.133 | 0.299 | 0.139 |
| Merlin | 0.182 | 0.271 | 0.149 |
| CT-CHAT | 0.182 | **0.382** | 0.171 |
| Ours-8 | 0.192 | 0.269 | 0.165 |
| Ours-16 | **0.266** | 0.272 | **0.329** |

**Baselines.** Prior methods rely on lightweight encoders and contrastive pretraining, limiting tokenization quality to capture relevant details for report generation. In contrast, our pretraining (with architectural and training advances) produces richer representations. We benchmark BTB3D against three state-of-the-art report generation models. CT2Rep [10] uses a CT-ViT encoder and GPT-

Table 3: Report generation performance on CT-RATE. BTB3D outperforms prior methods in both metric types, with the higher compression variant achieving the highest F1 and BLEU scores.

| | Clinical Accuracy ↑ | | | | Natural Language Generation ↑ | | | | | |
|---|---|---|---|---|---|---|---|---|---|---|
| Model | F1 | P | R | CRG | $B_1$ | $B_2$ | $B_3$ | $B_4$ | $B_{mean}$ | M |
| CT2Rep | 0.160 | 0.435 | 0.128 | 0.359 | 0.372 | 0.292 | 0.243 | 0.213 | 0.280 | 0.197 |
| Merlin | 0.160 | 0.295 | 0.112 | 0.352 | 0.231 | 0.163 | 0.124 | 0.099 | 0.154 | 0.148 |
| CT-CHAT | 0.184 | **0.450** | 0.158 | 0.368 | 0.373 | 0.284 | 0.231 | 0.198 | 0.272 | 0.215 |
| Ours-8 | 0.187 | 0.260 | 0.150 | 0.357 | 0.411 | 0.307 | 0.245 | **0.215** | 0.295 | 0.220 |
| Ours-16 | **0.258** | 0.260 | **0.260** | **0.370** | **0.439** | **0.320** | **0.248** | 0.213 | **0.305** | **0.223** |

Table 4: Text-conditional CT generation results. BTB3D with lower compression outperforms previous methods across all metrics, showing superior consistency, image quality, and text alignment.

| | FID ↓ | | | | FVD ↓ | | CLIP Score ↑ | |
|---|---|---|---|---|---|---|---|---|
| Model | Axial | Sagittal | Coronal | Mean | CT-Net | I3D | Text-Img | Img-Img |
| GenerateCT | 10.416 | 10.365 | 7.754 | 9.512 | 7.659 | 1512.5 | 23.625 | 84.287 |
| MedSyn | 14.963 | 12.115 | 10.698 | 12.592 | 13.927 | 725.81 | 23.571 | 84.153 |
| Ours-8 | **2.479** | **2.166** | **2.062** | **2.236** | **3.955** | **325.51** | **24.270** | **88.352** |
| Ours-16 | 7.077 | 4.226 | 3.729 | 5.011 | 4.020 | 429.34 | 23.322 | 84.957 |

style decoder with memory modules. CT-CHAT [3] aligns a CLIP-pretrained vision encoder with LLaMA [51]. Merlin [11] employs an I3D-ResNet backbone pretrained with masked and contrastive objectives and attaches a decoder similar to CT-CHAT. We use official weights for CT2Rep and CT-CHAT, both trained on CT-RATE. As Merlin was originally trained on private data and does not release weights, we retrain it on CT-RATE using the official codebase.

**Experimental results.** Table 3 reports clinical accuracy (F1, precision, recall, CRG [52]) and language quality (BLEU, METEOR [53, 54]) on CT-RATE. Merlin achieves high precision but low recall, tending to under-report findings, while CT2Rep and CT-CHAT show high recall but lower precision, often hallucinating abnormalities. Our method achieves a better balance, reflected in superior F1 and CRG scores. The $16 \times 16 \times 8$ variant achieves the highest F1, with a 40% relative improvement over CT-CHAT, confirming the effectiveness of our volumetric tokenization. Compared to CT-CHAT, BLEU-1 improves by 18% and BLEU-mean by 12%. Table 2 presents results on RadChestCT. Our $16 \times 16 \times 8$ model again achieves the highest F1 score, a 46% relative improvement over the best baselines. Recall also improves substantially, indicating strong generalization to out-of-distribution data. All clinical metrics are computed using the official CT-RATE report classifier.

**Dataset and implementation.** We use the CT-RATE dataset for report generation training, with the findings and impression sections used as in prior baselines. Chest CT scans are resampled to a voxel spacing of $0.75 \times 0.75 \times 1.5$ mm and padded or cropped to a uniform size of $512 \times 512 \times 241$. For external validation, we use RadChestCT [55], which contains multi-label annotations but no text reports. Models are trained for 40,000 iterations using DeepSpeed ZeRO-3 on 40 NVIDIA H100 GPUs. We use the AdamW optimizer with a learning rate of $2 \times 10^{-5}$ and a warm-up ratio of 0.03. Each GPU processes one sample, yielding an effective global batch size of 40. For LoRA, we set $r = 64, \alpha = 128$ for the $8^3$ variant, and $r = 128, \alpha = 256$ for the $16^2 \times 8$ variant.

## 4.3 Text-conditional 3D chest CT generation

We evaluate BTB3D's encoder-decoder on generating 3D chest CT scans from free-text prompts, assessing the realism, anatomical coherence, and text alignment of its latent representations. For generation, we use a transformer-based model with cross-attention layers, similar to GenerateCT [12].

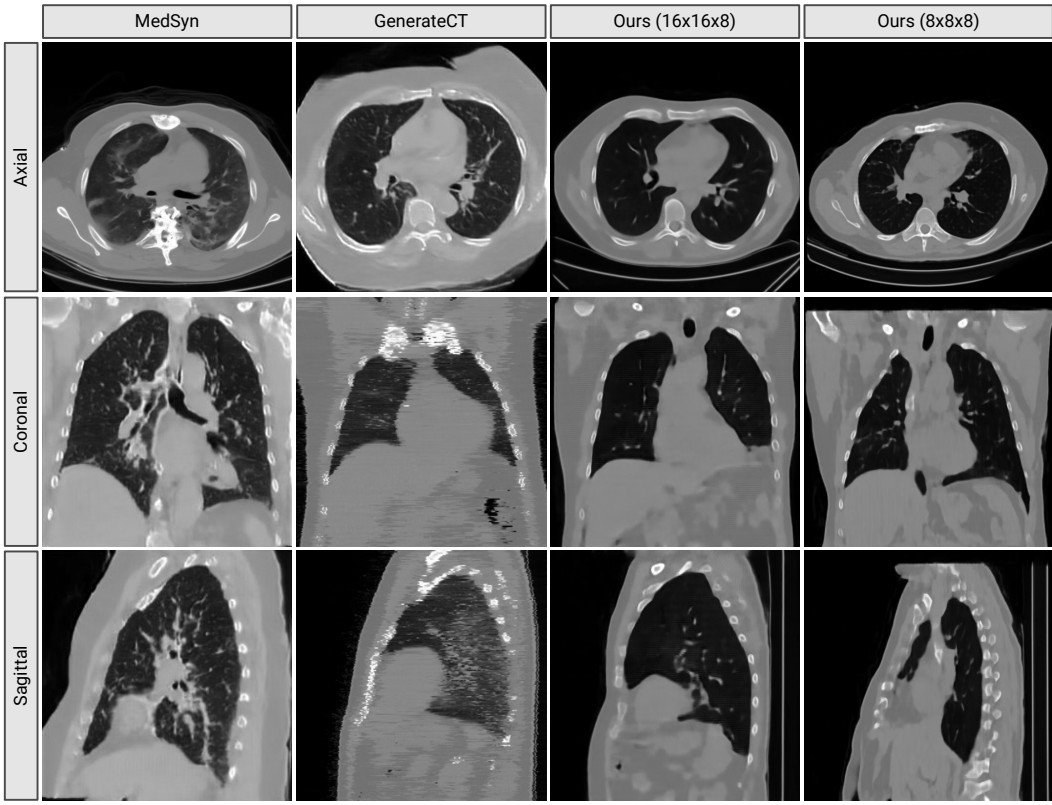

|  | MedSyn | GenerateCT | Ours (16x16x8) | Ours (8x8x8) |
|---|---|---|---|---|
| Axial | | | | |
| Coronal | | | | |
| Sagittal | | | | |

Figure 3: Example generations for a clinical prompt: *"Chest CT scan of a 63-year-old male: No findings compatible with pneumonia were detected. Mild sequelae changes are observed in both lungs. Atherosclerotic changes noted, with slight increase in the calibration of vascular structures in the mediastinum. Hepatosteatosis. Hiatal hernia."*. We show one representative slice per anatomical plane. Ground-truth volumes are omitted, following standard practice in text-to-image generation.

**Baselines.** GenerateCT [12] adopts a cascaded design (low-resolution 3D generation followed by 2D diffusion upsampling), which leads to spatial inconsistencies. This approach is motivated by the computational constraints of training a high-capacity 3D encoder-decoder directly on high-resolution volumes. MedSyn [20] employs a unified 3D transformer with windowed attention to improve inter-slice consistency but suffers from lower fidelity due to its lightweight architecture.

**Experimental results.** Table 4 reports FID (per view and mean), FVD (with CT-Net [55] and I3D backbones), and CLIP-based alignment scores. Since CT-Net is trained on 3D chest CT volumes, it provides a more relevant assessment than I3D. BTB3D with lower compression achieves substantial improvements: mean FID drops from 9.51 (GenerateCT) to 2.24 (a 76.5% reduction). $FVD_{CT-Net}$ improves by 48.3%, confirming better spatiotemporal realism, while CLIP alignment also slightly increases. The higher-compression BTB3D variant still outperforms GenerateCT and MedSyn across all metrics, though with smaller margins, highlighting a trade-off between compression and generation quality. These results underscore the effectiveness of our encoder-decoder network for high-fidelity, text-aligned 3D medical image synthesis. Figure 3 illustrates qualitative differences: our model generates sharper volumes with clearer anatomical structures and better alignment to the clinical prompt, while MedSyn and GenerateCT outputs appear blurrier or contain inter-slice artifacts.

**Dataset and implementation.** We use CT-RATE, with CTs resampled as in Section 4.2. Prompts are generated as: *"Chest CT scan of a {age}-year-old {sex}: {impression}"*, following GenerateCT. The generation model is a 12-layer transformer (1024 hidden size, 16 heads), trained using flow matching loss [56]. We apply $[7, 7, 7]$ windowed self-attention and $[2, 2, 2]$ patching. Prompts are encoded using the T5v1.1-base model [57]. Training is conducted on 16 H100 GPUs for 1500 epochs, using AdamW with a learning rate of $10^{-4}$. The batch size is 4 for the $8^3$ variant and 8 for $16^2 \times 8$.

# 5    Conclusion

We introduced BTB3D, a framework that advances vision-language modeling for 3D medical imaging. It unifies 2D and 3D processing through a causal convolutional encoder-decoder and compact volumetric tokenization. Our three-stage training scales learning from local patterns to full-volume anatomical coherence, addressing memory and resolution bottlenecks. BTB3D achieves state-of-the-art results in both radiology report generation and text-conditioned 3D CT synthesis. Notably, the $8{\times}8{\times}8$ variant excels in fine-grained tasks such as text-to-CT synthesis, while the $16{\times}16{\times}8$ variant is better suited for memory-constrained settings and high-level tasks like report generation. We believe BTB3D is a significant step toward scalable and clinically meaningful vision-language modeling in 3D medical imaging, and we expect its open-source release to catalyze further research.

## Acknowledgments and Disclosure of Funding

High-performance computing resources were provided by the Erlangen National High Performance Computing Center (NHR@FAU) at Friedrich-Alexander-Universität Erlangen-Nürnberg (FAU), under the NHR projects b143dc and b180dc. NHR is funded by federal and Bavarian state authorities, and NHR@FAU hardware is partially funded by the German Research Foundation (DFG) – 440719683. Additional support was received by the ERC - project MIA-NORMAL 101083647, DFG 512819079, and by the state of Bavaria (HTA). The authors also express sincere gratitude to the Helmut-Horten Foundation for their generous support, which made this work possible.

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

# NeurIPS Paper Checklist

1. **Claims**

   Question: Do the main claims made in the abstract and introduction accurately reflect the paper's contributions and scope?

   Answer: [Yes]

   Justification: The abstract and introduction clearly state two main claims: (1) BTB3D achieves state-of-the-art performance on both radiology report generation and text-conditional 3D CT synthesis, and (2) its improvements stem from a unified causal encoder-decoder architecture and a three-stage training strategy. These claims are thoroughly validated in Section 4 through reconstruction analysis (Section 4.1), report generation benchmarks (Section 4.2), and CT synthesis metrics (Section 4.3). Quantitative improvements (such as a 40% F1 gain over CT-CHAT for report generation and a 76.5% FID reduction over GenerateCT) directly support the claims. The paper also discusses architectural trade-offs and generalization, aligning the claims with the demonstrated scope and limitations.

   Guidelines:

   - The answer NA means that the abstract and introduction do not include the claims made in the paper.
   - The abstract and/or introduction should clearly state the claims made, including the contributions made in the paper and important assumptions and limitations. A No or NA answer to this question will not be perceived well by the reviewers.
   - The claims made should match theoretical and experimental results, and reflect how much the results can be expected to generalize to other settings.
   - It is fine to include aspirational goals as motivation as long as it is clear that these goals are not attained by the paper.

2. **Limitations**

   Question: Does the paper discuss the limitations of the work performed by the authors?

   Answer: [Yes]

   Justification: The paper discusses several limitations throughout and includes a dedicated section in the Appendix. Specifically, we acknowledge that BTB3D is evaluated only on chest CTs due to the lack of large-scale paired datasets for other anatomies and modalities. Also, our method requires substantial computational resources for training, which may limit accessibility. Finally, although we evaluate performance using both clinical and generative metrics, further validation through expert reader studies and prospective clinical trials is necessary to assess real-world safety and efficacy.

   Guidelines:

   - The answer NA means that the paper has no limitation while the answer No means that the paper has limitations, but those are not discussed in the paper.
   - The authors are encouraged to create a separate "Limitations" section in their paper.
   - The paper should point out any strong assumptions and how robust the results are to violations of these assumptions (e.g., independence assumptions, noiseless settings, model well-specification, asymptotic approximations only holding locally). The authors should reflect on how these assumptions might be violated in practice and what the implications would be.
   - The authors should reflect on the scope of the claims made, e.g., if the approach was only tested on a few datasets or with a few runs. In general, empirical results often depend on implicit assumptions, which should be articulated.
   - The authors should reflect on the factors that influence the performance of the approach. For example, a facial recognition algorithm may perform poorly when image resolution is low or images are taken in low lighting. Or a speech-to-text system might not be used reliably to provide closed captions for online lectures because it fails to handle technical jargon.
   - The authors should discuss the computational efficiency of the proposed algorithms and how they scale with dataset size.

- If applicable, the authors should discuss possible limitations of their approach to address problems of privacy and fairness.
- While the authors might fear that complete honesty about limitations might be used by reviewers as grounds for rejection, a worse outcome might be that reviewers discover limitations that aren't acknowledged in the paper. The authors should use their best judgment and recognize that individual actions in favor of transparency play an important role in developing norms that preserve the integrity of the community. Reviewers will be specifically instructed to not penalize honesty concerning limitations.

3. **Theory assumptions and proofs**

Question: For each theoretical result, does the paper provide the full set of assumptions and a complete (and correct) proof?

Answer: [NA]

Justification: The paper is focused on an empirical vision-language modeling framework for 3D medical imaging and does not contain theoretical results, assumptions, or proofs.

Guidelines:

- The answer NA means that the paper does not include theoretical results.
- All the theorems, formulas, and proofs in the paper should be numbered and cross-referenced.
- All assumptions should be clearly stated or referenced in the statement of any theorems.
- The proofs can either appear in the main paper or the supplemental material, but if they appear in the supplemental material, the authors are encouraged to provide a short proof sketch to provide intuition.
- Inversely, any informal proof provided in the core of the paper should be complemented by formal proofs provided in appendix or supplemental material.
- Theorems and Lemmas that the proof relies upon should be properly referenced.

4. **Experimental result reproducibility**

Question: Does the paper fully disclose all the information needed to reproduce the main experimental results of the paper to the extent that it affects the main claims and/or conclusions of the paper (regardless of whether the code and data are provided or not)?

Answer: [Yes]

Justification: We provide detailed methodology in Section 3 and implementation details in Section 4. Our experiments use the publicly available CT-RATE dataset for training and in-domain evaluation, and RadChestCT for out-of-distribution validation. We release all training and validation scripts, along with pretrained model weights, to ensure full reproducibility of the results and support verification of all main claims.

Guidelines:

- The answer NA means that the paper does not include experiments.
- If the paper includes experiments, a No answer to this question will not be perceived well by the reviewers: Making the paper reproducible is important, regardless of whether the code and data are provided or not.
- If the contribution is a dataset and/or model, the authors should describe the steps taken to make their results reproducible or verifiable.
- Depending on the contribution, reproducibility can be accomplished in various ways. For example, if the contribution is a novel architecture, describing the architecture fully might suffice, or if the contribution is a specific model and empirical evaluation, it may be necessary to either make it possible for others to replicate the model with the same dataset, or provide access to the model. In general. releasing code and data is often one good way to accomplish this, but reproducibility can also be provided via detailed instructions for how to replicate the results, access to a hosted model (e.g., in the case of a large language model), releasing of a model checkpoint, or other means that are appropriate to the research performed.
- While NeurIPS does not require releasing code, the conference does require all submissions to provide some reasonable avenue for reproducibility, which may depend on the nature of the contribution. For example

(a) If the contribution is primarily a new algorithm, the paper should make it clear how to reproduce that algorithm.

(b) If the contribution is primarily a new model architecture, the paper should describe the architecture clearly and fully.

(c) If the contribution is a new model (e.g., a large language model), then there should either be a way to access this model for reproducing the results or a way to reproduce the model (e.g., with an open-source dataset or instructions for how to construct the dataset).

(d) We recognize that reproducibility may be tricky in some cases, in which case authors are welcome to describe the particular way they provide for reproducibility. In the case of closed-source models, it may be that access to the model is limited in some way (e.g., to registered users), but it should be possible for other researchers to have some path to reproducing or verifying the results.

5. **Open access to data and code**

Question: Does the paper provide open access to the data and code, with sufficient instructions to faithfully reproduce the main experimental results, as described in supplemental material?

Answer: [Yes]

Justification: We will open-source our codebase, including all training, validation, and evaluation scripts. The repository will contain detailed instructions on setting up the environment, accessing and preprocessing the CT-RATE and RadChestCT datasets, and reproducing all experiments reported in the paper. Pretrained weights for BTB3D and the generation models will also be provided to ensure faithful replication of our results.

Guidelines:

- The answer NA means that paper does not include experiments requiring code.
- Please see the NeurIPS code and data submission guidelines (`https://nips.cc/public/guides/CodeSubmissionPolicy`) for more details.
- While we encourage the release of code and data, we understand that this might not be possible, so "No" is an acceptable answer. Papers cannot be rejected simply for not including code, unless this is central to the contribution (e.g., for a new open-source benchmark).
- The instructions should contain the exact command and environment needed to run to reproduce the results. See the NeurIPS code and data submission guidelines (`https://nips.cc/public/guides/CodeSubmissionPolicy`) for more details.
- The authors should provide instructions on data access and preparation, including how to access the raw data, preprocessed data, intermediate data, and generated data, etc.
- The authors should provide scripts to reproduce all experimental results for the new proposed method and baselines. If only a subset of experiments are reproducible, they should state which ones are omitted from the script and why.
- At submission time, to preserve anonymity, the authors should release anonymized versions (if applicable).
- Providing as much information as possible in supplemental material (appended to the paper) is recommended, but including URLs to data and code is permitted.

6. **Experimental setting/details**

Question: Does the paper specify all the training and test details (e.g., data splits, hyper-parameters, how they were chosen, type of optimizer, etc.) necessary to understand the results?

Answer: [Yes]

Justification: All training and evaluation details are thoroughly described in Section 4, including dataset splits, preprocessing steps, model configurations, optimizer settings, batch sizes, learning rates, training schedules, and hardware setup. We also specify stage-wise training parameters and evaluation protocols for each task, ensuring the experimental setup is fully transparent and reproducible.

Guidelines:

- The answer NA means that the paper does not include experiments.
- The experimental setting should be presented in the core of the paper to a level of detail that is necessary to appreciate the results and make sense of them.
- The full details can be provided either with the code, in appendix, or as supplemental material.

7. **Experiment statistical significance**

   Question: Does the paper report error bars suitably and correctly defined or other appropriate information about the statistical significance of the experiments?

   Answer: [No]

   Justification: Due to the substantial computational cost of training large 3D models on high-resolution CT volumes, we did not conduct repeated runs or report statistical significance metrics such as error bars or confidence intervals. Our focus was on demonstrating consistent performance gains across multiple benchmarks using standardized evaluation protocols. We acknowledge this as a limitation and will consider statistical analysis in future work.

   Guidelines:

   - The answer NA means that the paper does not include experiments.
   - The authors should answer "Yes" if the results are accompanied by error bars, confidence intervals, or statistical significance tests, at least for the experiments that support the main claims of the paper.
   - The factors of variability that the error bars are capturing should be clearly stated (for example, train/test split, initialization, random drawing of some parameter, or overall run with given experimental conditions).
   - The method for calculating the error bars should be explained (closed form formula, call to a library function, bootstrap, etc.)
   - The assumptions made should be given (e.g., Normally distributed errors).
   - It should be clear whether the error bar is the standard deviation or the standard error of the mean.
   - It is OK to report 1-sigma error bars, but one should state it. The authors should preferably report a 2-sigma error bar than state that they have a 96% CI, if the hypothesis of Normality of errors is not verified.
   - For asymmetric distributions, the authors should be careful not to show in tables or figures symmetric error bars that would yield results that are out of range (e.g. negative error rates).
   - If error bars are reported in tables or plots, The authors should explain in the text how they were calculated and reference the corresponding figures or tables in the text.

8. **Experiments compute resources**

   Question: For each experiment, does the paper provide sufficient information on the computer resources (type of compute workers, memory, time of execution) needed to reproduce the experiments?

   Answer: [Yes]

   Justification: We detail the hardware and software setup in Section 4. Batch sizes, memory constraints, and stage-wise training configurations are specified, allowing estimation of training time and resource requirements. This enables reproducibility and helps assess the computational cost of our method.

   Guidelines:

   - The answer NA means that the paper does not include experiments.
   - The paper should indicate the type of compute workers CPU or GPU, internal cluster, or cloud provider, including relevant memory and storage.
   - The paper should provide the amount of compute required for each of the individual experimental runs as well as estimate the total compute.
   - The paper should disclose whether the full research project required more compute than the experiments reported in the paper (e.g., preliminary or failed experiments that didn't make it into the paper).

9. **Code of ethics**

   Question: Does the research conducted in the paper conform, in every respect, with the NeurIPS Code of Ethics https://neurips.cc/public/EthicsGuidelines?

   Answer: [Yes]

   Justification: We have carefully reviewed the NeurIPS Code of Ethics and ensured that all aspects of our research (including data usage, model development, evaluation, and reporting) fully comply with its principles. The datasets used are publicly available, de-identified, and ethically sourced, and we maintain transparency and reproducibility throughout the work.

   Guidelines:

   - The answer NA means that the authors have not reviewed the NeurIPS Code of Ethics.
   - If the authors answer No, they should explain the special circumstances that require a deviation from the Code of Ethics.
   - The authors should make sure to preserve anonymity (e.g., if there is a special consideration due to laws or regulations in their jurisdiction).

10. **Broader impacts**

    Question: Does the paper discuss both potential positive societal impacts and negative societal impacts of the work performed?

    Answer: [Yes]

    Justification: We provide a discussion of both potential positive and negative societal impacts in the Appendix. On the positive side, BTB3D has the potential to improve diagnostic accuracy, reduce radiologist workload, and enable better access to training tools through high-quality synthetic CT data. On the negative side, we acknowledge potential risks such as misuse of generative models for medical fraud or reconstruction of sensitive information from improperly anonymized data. We emphasize the importance of proper anonymization and include cautionary notes regarding ethical deployment and data governance.

    Guidelines:

    - The answer NA means that there is no societal impact of the work performed.
    - If the authors answer NA or No, they should explain why their work has no societal impact or why the paper does not address societal impact.
    - Examples of negative societal impacts include potential malicious or unintended uses (e.g., disinformation, generating fake profiles, surveillance), fairness considerations (e.g., deployment of technologies that could make decisions that unfairly impact specific groups), privacy considerations, and security considerations.
    - The conference expects that many papers will be foundational research and not tied to particular applications, let alone deployments. However, if there is a direct path to any negative applications, the authors should point it out. For example, it is legitimate to point out that an improvement in the quality of generative models could be used to generate deepfakes for disinformation. On the other hand, it is not needed to point out that a generic algorithm for optimizing neural networks could enable people to train models that generate Deepfakes faster.
    - The authors should consider possible harms that could arise when the technology is being used as intended and functioning correctly, harms that could arise when the technology is being used as intended but gives incorrect results, and harms following from (intentional or unintentional) misuse of the technology.
    - If there are negative societal impacts, the authors could also discuss possible mitigation strategies (e.g., gated release of models, providing defenses in addition to attacks, mechanisms for monitoring misuse, mechanisms to monitor how a system learns from feedback over time, improving the efficiency and accessibility of ML).

11. **Safeguards**

    Question: Does the paper describe safeguards that have been put in place for responsible release of data or models that have a high risk for misuse (e.g., pretrained language models, image generators, or scraped datasets)?

    Answer: [Yes]

Justification: While we rely solely on publicly available datasets, we acknowledge the potential misuse of generative models. To mitigate this, access to our pretrained models is gated: researchers must apply for access to the model weights, and requests are manually reviewed to ensure responsible use. This safeguard helps prevent fraudulent or unethical deployment of our generation models.

Guidelines:

- The answer NA means that the paper poses no such risks.
- Released models that have a high risk for misuse or dual-use should be released with necessary safeguards to allow for controlled use of the model, for example by requiring that users adhere to usage guidelines or restrictions to access the model or implementing safety filters.
- Datasets that have been scraped from the Internet could pose safety risks. The authors should describe how they avoided releasing unsafe images.
- We recognize that providing effective safeguards is challenging, and many papers do not require this, but we encourage authors to take this into account and make a best faith effort.

12. **Licenses for existing assets**

Question: Are the creators or original owners of assets (e.g., code, data, models), used in the paper, properly credited and are the license and terms of use explicitly mentioned and properly respected?

Answer: [Yes]

Justification: All external assets used are publicly available datasets, and we comply with their respective licenses (We cite the original papers and follow license terms). Specifically:

- **CT-RATE dataset:** Released under the CC BY-NC-SA 4.0 license.
- **RadChestCT dataset:** Released under the CC BY-NC-ND 4.0 license.

Guidelines:

- The answer NA means that the paper does not use existing assets.
- The authors should cite the original paper that produced the code package or dataset.
- The authors should state which version of the asset is used and, if possible, include a URL.
- The name of the license (e.g., CC-BY 4.0) should be included for each asset.
- For scraped data from a particular source (e.g., website), the copyright and terms of service of that source should be provided.
- If assets are released, the license, copyright information, and terms of use in the package should be provided. For popular datasets, `paperswithcode.com/datasets` has curated licenses for some datasets. Their licensing guide can help determine the license of a dataset.
- For existing datasets that are re-packaged, both the original license and the license of the derived asset (if it has changed) should be provided.
- If this information is not available online, the authors are encouraged to reach out to the asset's creators.

13. **New assets**

Question: Are new assets introduced in the paper well documented and is the documentation provided alongside the assets?

Answer: [Yes]

Justification: We will release all new assets (including code, pretrained models, and training scripts) under the CC BY 4.0 license. The GitHub repository includes comprehensive documentation covering usage instructions, training configurations, dependencies, and licensing terms to facilitate reproducibility and reuse.

Guidelines:

- The answer NA means that the paper does not release new assets.

- Researchers should communicate the details of the dataset/code/model as part of their submissions via structured templates. This includes details about training, license, limitations, etc.
- The paper should discuss whether and how consent was obtained from people whose asset is used.
- At submission time, remember to anonymize your assets (if applicable). You can either create an anonymized URL or include an anonymized zip file.

14. **Crowdsourcing and research with human subjects**

Question: For crowdsourcing experiments and research with human subjects, does the paper include the full text of instructions given to participants and screenshots, if applicable, as well as details about compensation (if any)?

Answer: [NA]

Justification: This work does not involve crowdsourcing or research with human subjects.

Guidelines:

- The answer NA means that the paper does not involve crowdsourcing nor research with human subjects.
- Including this information in the supplemental material is fine, but if the main contribution of the paper involves human subjects, then as much detail as possible should be included in the main paper.
- According to the NeurIPS Code of Ethics, workers involved in data collection, curation, or other labor should be paid at least the minimum wage in the country of the data collector.

15. **Institutional review board (IRB) approvals or equivalent for research with human subjects**

Question: Does the paper describe potential risks incurred by study participants, whether such risks were disclosed to the subjects, and whether Institutional Review Board (IRB) approvals (or an equivalent approval/review based on the requirements of your country or institution) were obtained?

Answer: [NA]

Justification: This work does not involve human subjects and therefore does not require IRB.

Guidelines:

- The answer NA means that the paper does not involve crowdsourcing nor research with human subjects.
- Depending on the country in which research is conducted, IRB approval (or equivalent) may be required for any human subjects research. If you obtained IRB approval, you should clearly state this in the paper.
- We recognize that the procedures for this may vary significantly between institutions and locations, and we expect authors to adhere to the NeurIPS Code of Ethics and the guidelines for their institution.
- For initial submissions, do not include any information that would break anonymity (if applicable), such as the institution conducting the review.

16. **Declaration of LLM usage**

Question: Does the paper describe the usage of LLMs if it is an important, original, or non-standard component of the core methods in this research? Note that if the LLM is used only for writing, editing, or formatting purposes and does not impact the core methodology, scientific rigorousness, or originality of the research, declaration is not required.

Answer: [NA]

Justification: LLMs were used only for writing and editing assistance, not as part of the core methodology.

Guidelines:

- The answer NA means that the core method development in this research does not involve LLMs as any important, original, or non-standard components.
- Please refer to our LLM policy (`https://neurips.cc/Conferences/2025/LLM`) for what should or should not be described.

# A  Technical Appendices and Supplementary Material

In this supplementary document, we provide additional insights and supporting materials for our main paper. We begin by outlining the key limitations of our work and discussing the broader societal and clinical implications of the BTB3D framework, particularly with regard to its capabilities in radiology report generation and text-conditioned 3D CT volume synthesis. We then present further experimental details that expand upon the methodology described in the main text. Additional qualitative and quantitative results are also included to further validate our findings. All code, pretrained model weights, and instructions to reproduce our experiments will be made openly available in our GitHub repository to promote transparency, reproducibility, and further research in the field.

## A.1  Limitations and Broader Impacts

**Limitations.**  While our BTB3D framework significantly improves tokenization and decoding for 3D medical vision-language modeling (especially for 3D chest CT scans), several limitations remain. First, our current framework does not explicitly model clinical reasoning or uncertainty, both of which are crucial for real-world deployment in clinical settings. Incorporating modules for uncertainty estimation or causal inference remains an open challenge for future studies.

Second, the scope of our experiments is limited to 3D chest CT scans, primarily due to the lack of large-scale, publicly available paired datasets (with reports) for other anatomical regions or modalities (*e.g.*, MRI, PET). While BTB3D is designed to generalize across 2D and 3D inputs, we have not yet validated its transferability to other clinical domains. Future efforts should extend the architecture to whole-body imaging and explore zero-shot or few-shot generalization across organs.

Third, although BTB3D supports both 2D and 3D training modes, we used only 2D slices extracted from 3D volumes in CT-RATE to remain consistent with baselines and ensure fair comparisons. As such, we have not demonstrated its transfer capabilities (*e.g.*, transfer from pretrained 2D models or joint training with 2D paired datasets), which limits the evaluation of BTB3D's capabilities.

Fourth, statistical analysis is limited due to the high computational cost of 3D training. We did not perform repeated runs or report confidence intervals. Thus, while our results demonstrate strong and consistent performance across multiple benchmarks, statistical significance remains to be validated.

Lastly, although we evaluate both clinical and generative metrics, BTB3D has not yet been assessed in real clinical workflows. Validation through expert reader studies or prospective trials is necessary to fully establish its utility, reliability, and safety in decision-making environments.

**Broader impacts.**  BTB3D offers promising benefits for both clinical and research communities. High-quality radiology report generation from 3D CT scans can reduce reporting delays, alleviate radiologist burnout, and improve documentation consistency, particularly in high-volume settings such as emergency departments or large-scale screening programs. Moreover, BTB3D's ability to synthesize realistic, anatomically plausible CT scans from text opens new avenues for training, simulation, and rare disease modeling. In educational settings, synthetic CT scans can be used to create diverse training sets for radiology students and enhance learning outcomes through exposure to rare or atypical cases. In research, BTB3D may support data augmentation, helping mitigate class imbalance in supervised learning tasks and enabling pretraining in low-resource environments.

However, the generative capabilities of the BTB3D framework also present potential risks. Synthetic scans, if not properly labeled or constrained, may be misused in regulatory or insurance contexts, or exploited for fraudulent purposes. Additionally, there is a non-trivial risk of inadvertently replicating identifiable patient anatomy, even when training data is anonymized. While our work uses only publicly available, fully de-identified datasets (*e.g.*, CT-RATE), developers must ensure strict compliance with data governance and anonymization protocols when deploying similar models.

Finally, the substantial computational cost of training such models may exacerbate disparities in access to advanced medical AI. Ensuring open-source availability and supporting low-resource inference are critical to democratizing BTB3D's benefits. Continued oversight and interdisciplinary dialogue will be essential to ensuring the responsible deployment of generative models in healthcare.

## A.2 Additional Experimental Details

**Three-stage training performance.** To further evaluate the impact of our progressive training strategy, we qualitatively assess the reconstruction performance of BTB3D across the three stages of training in Figure 4. The figure presents axial, coronal, and sagittal slices reconstructed by our two model variants ($16{\times}16{\times}8$ and $8{\times}8{\times}8$) at each stage and compares them to the ground truth.

In *Stage 1*, the model is trained solely on small, non-overlapping 3D volumes and 2D slices. This initialization allows the model to learn basic volumetric structure and inter-slice continuity but suffers from block artifacts and poor spatial coherence, especially in regions with high anatomical complexity. As seen in the leftmost columns of Figure 4, both variants at this stage produce noisy and blurry reconstructions with limited detail and sharpness. *Stage 2* introduces overlapping windows during training, which significantly improves anatomical consistency across slices and helps reduce the block discontinuities learned in Stage 1. This stage yields the largest qualitative leap in fidelity, with more well-defined boundaries of lung lobes, airways, and soft tissue structures. The improvements are most noticeable in the axial and sagittal views, where inter-slice alignment and smoothness are critical for clinical interpretability. *Stage 3* refines the decoder with high-resolution patches while keeping the encoder part frozen. This final stage enhances local detail, texture realism, and structural sharpness without sacrificing the global consistency achieved in Stage 2. Notably, lung fissures, pleural contours, and fine vascular structures become visibly clearer, indicating that the decoder has learned to reconstruct complex anatomical regions with higher fidelity and resolution.

Comparing the two BTB3D variants, the $8{\times}8{\times}8$ model consistently achieves superior anatomical fidelity and inter-slice coherence in reconstruction, as expected due to its lower compression rate and higher capacity. In contrast, the $16{\times}16{\times}8$ variant (while more compressed) proves more effective for language-driven tasks such as report generation, where coarse global structure suffices and memory efficiency is critical. Our three-stage training pipeline plays a pivotal role for both models: it first captures global structure, then progressively refines local detail, enabling accurate and high-resolution 3D reconstructions from compact tokens. This staged optimization bridges the gap between token efficiency and clinical utility, facilitating both high-fidelity text-conditional CT generation and precise report generation, underscoring BTB3D's versatility in multimodal 3D medical image understanding.

**Radiology report generation from 3D chest CT.** We comprehensively evaluate BTB3D's capability in generating accurate and clinically coherent radiology reports from volumetric chest CT scans. As described in the main paper, each CT volume is first compressed into a compact sequence of frequency-aware 3D tokens by our encoder, and these tokens are then passed to a pretrained LLM (LLaMA 3.1 8B) for report generation via a linear projection layer. To ensure fairness, we use the official weights for CT2Rep and CT-CHAT, both trained on the same CT-RATE dataset. Since Merlin was originally trained on a private dataset and neither its weights nor training data are publicly available, we retrained Merlin on CT-RATE using its official codebase and default hyperparameters.

Our evaluation includes both internal (CT-RATE test set) and external (RAD-ChestCT) benchmarks. The results in Table 5 show that BTB3D (particularly the $16{\times}16{\times}8$ variant) outperforms prior methods in average abnormality level F1 scores, achieving higher clinical precision in most of the findings. This trend holds in Table 6, where our BTB3D framework demonstrates strong out-of-distribution generalization, with the $16{\times}16{\times}8$ variant yielding a 46% relative F1 improvement over CT-CHAT (the previous state-of-the-art method). This highlights the robustness of our tokenization and training pipeline, even when evaluated on unseen institutional distributions.

Figure 5 offers a qualitative comparison across models, illustrating report generation for the same CT scan. BTB3D's reports more faithfully reproduce key clinical details from the ground truth, particularly with the higher compression ($16{\times}16{\times}8$) variant, supporting the notion that coarser representations, while less suitable for pixel-level synthesis, may better capture global semantics for language modeling. Meanwhile, the $8{\times}8{\times}8$ variant provides more spatially detailed reconstructions but slightly underperforms on text generation metrics, suggesting a trade-off between compression depth and semantic abstraction. In all comparisons, BTB3D demonstrates a compelling advantage by combining precise volumetric tokenization with a scalable training strategy, ultimately allowing both fine-grained anatomical reconstruction and clinically relevant report generation.

**Text-conditional 3D chest CT generation.**    To evaluate the generative capabilities of BTB3D, we benchmark its performance on synthesizing realistic and anatomically coherent 3D chest CT volumes from free-text clinical prompts. As shown in Figure 6, BTB3D generates sharper, more anatomically faithful volumes compared to MedSyn and GenerateCT, with improved inter-slice consistency and alignment to prompt semantics. For fair comparison, we use the official pretrained weights for both MedSyn and GenerateCT, which were trained on the same modality (3D chest CT), ensuring that differences in performance stem from architectural and training innovations.

Quantitative results reported in the main paper (Table 4) show that our BTB3D framework with the lower compression variant ($8{\times}8{\times}8$) achieves the best overall performance across all generative metrics. Specifically, it reduces the mean Fréchet Inception Distance (FID) from 9.51 (GenerateCT) and 12.59 (MedSyn) to just 2.24, a 76.5% improvement, demonstrating superior fidelity. We compute FID using the `FIDMetric` from the MONAI library [58], leveraging a RadImageNet-pretrained ResNet50 backbone [59], which is better suited for grayscale radiology images than traditional Inception networks. Following standard practice established by GenerateCT and MAISI [24], FID is calculated on the central 40% of slices (in each anatomical plane) across 100 randomly selected volumes per method, reducing boundary noise and focusing on clinically relevant regions.

In terms of temporal and anatomical realism, Fréchet Video Distance (FVD) is computed using both CT-Net (specialized for 3D chest CTs) [55] and I3D (trained on RGB videos). BTB3D again significantly outperforms prior work, halving the FVD compared to GenerateCT. To assess semantic alignment between text prompts and generated volumes, we compute CLIP scores using the `CLIPScore` implementation from Torchmetrics. We follow GenerateCT's protocol: axial slices are resized to $224 \times 224$ and converted to pseudo-RGB by repeating the single intensity channel. Using `clip-vit-base-patch16`, we observe that BTB3D achieves the highest text-image alignment score (24.27), suggesting it captures fine semantic cues better than prior methods.

Interestingly, we note a compression-quality trade-off: while the $8{\times}8{\times}8$ variant excels in fine-grained reconstruction and text-conditional volume synthesis, the more compact $16{\times}16{\times}8$ variant still surpasses existing baselines and may be preferable in memory-constrained or latency-sensitive settings. Together, these results confirm that BTB3D's volumetric tokenization and three-stage training pipeline offer a significant leap forward in text-conditional 3D medical image generation, bridging the gap between semantic understanding and pixel-level anatomical coherence.

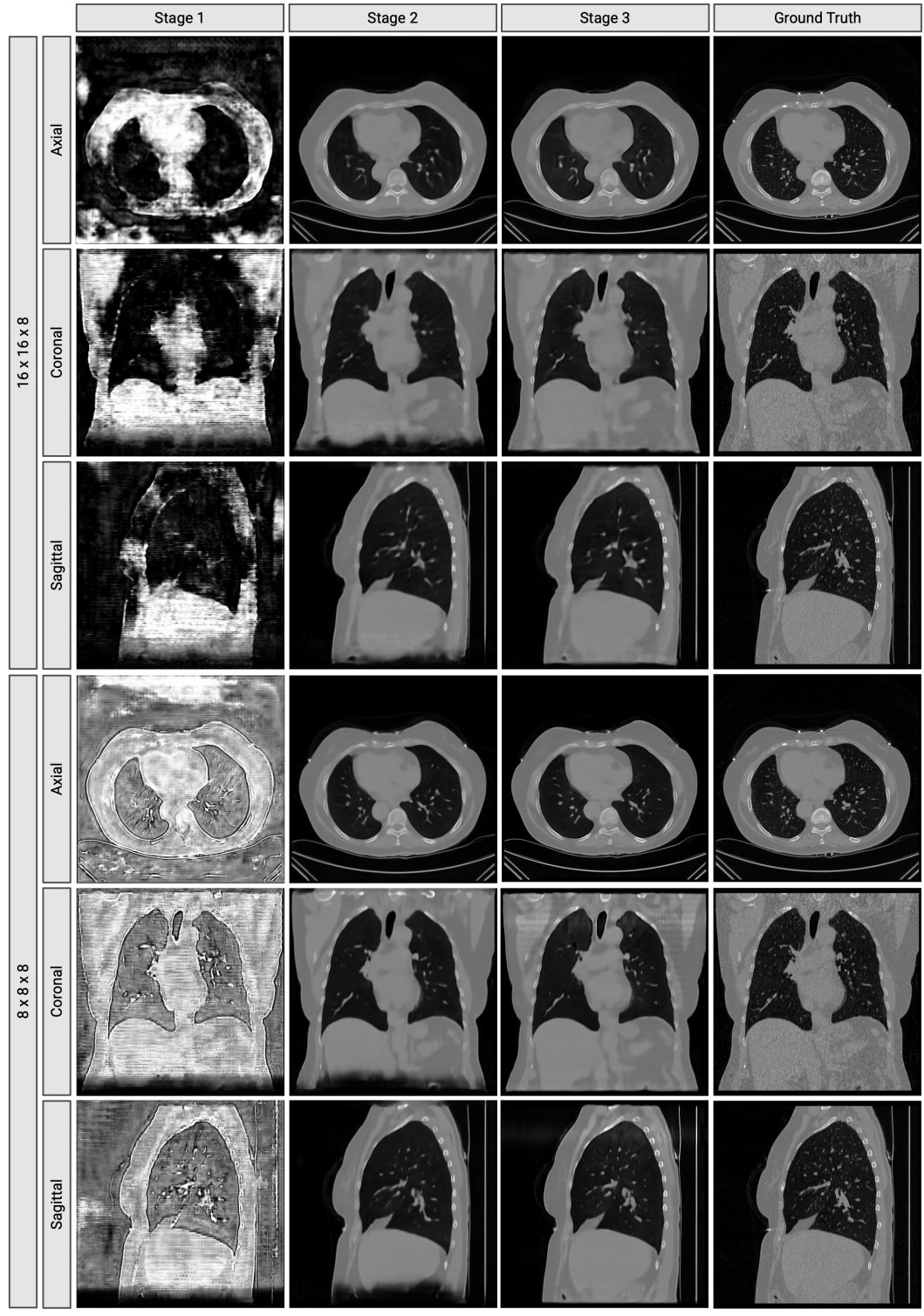

Figure 4: Qualitative reconstruction results across axial, coronal, and sagittal planes for two BTB3D variants: $16\times16\times8$ (top) and $8\times8\times8$ (bottom). The figure shows outputs after Stage 1 (short-volume training), Stage 2 (overlapping-window training), and Stage 3 (decoder refinement), compared to the ground truth. The progressive improvements highlight the effectiveness of our three-stage training strategy. Stage 2 yields the largest gain in anatomical fidelity and inter-slice consistency, while Stage 3 further sharpens structural details such as lung fissures and vascular boundaries.

Table 5: Abnormality-based F1 scores on CT-RATE along with the prevalence ratio of each abnormality in the test set. BTB3D consistently achieves the highest clinical accuracy across most categories, with the higher-compression variant ($16\times16\times8$) demonstrating superior overall performance.

| Abnormality | Ratio | Ours-16 | Ours-8 | CT-CHAT | Merlin | CT2Rep |
|---|---|---|---|---|---|---|
| Medical material | 0.103 | 0.142 | 0.120 | 0.006 | 0.057 | 0.000 |
| Arterial wall calcification | 0.285 | 0.414 | 0.273 | 0.451 | 0.262 | 0.322 |
| Cardiomegaly | 0.107 | 0.305 | 0.207 | 0.123 | 0.176 | 0.013 |
| Pericardial effusion | 0.074 | 0.095 | 0.095 | 0.009 | 0.060 | 0.000 |
| Coronary artery wall calc. | 0.252 | 0.403 | 0.260 | 0.412 | 0.235 | 0.335 |
| Hiatal hernia | 0.137 | 0.164 | 0.118 | 0.207 | 0.110 | 0.074 |
| Lymphadenopathy | 0.260 | 0.358 | 0.209 | 0.069 | 0.227 | 0.013 |
| Emphysema | 0.197 | 0.196 | 0.155 | 0.391 | 0.216 | 0.198 |
| Atelectasis | 0.235 | 0.269 | 0.242 | 0.341 | 0.199 | 0.323 |
| Lung nodule | 0.448 | 0.427 | 0.397 | 0.443 | 0.290 | 0.029 |
| Lung opacity | 0.390 | 0.408 | 0.382 | 0.266 | 0.312 | 0.557 |
| Pulmonary fibrotic sequela | 0.273 | 0.318 | 0.211 | 0.069 | 0.117 | 0.104 |
| Pleural effusion | 0.124 | 0.308 | 0.199 | 0.173 | 0.183 | 0.341 |
| Mosaic attenuation pattern | 0.083 | 0.183 | 0.094 | 0.064 | 0.076 | 0.198 |
| Peribronchial thickening | 0.117 | 0.125 | 0.043 | 0.000 | 0.054 | 0.099 |
| Consolidation | 0.191 | 0.259 | 0.185 | 0.120 | 0.174 | 0.236 |
| Bronchiectasis | 0.109 | 0.126 | 0.094 | 0.091 | 0.075 | 0.013 |
| Interlobular septal thick. | 0.082 | 0.135 | 0.087 | 0.075 | 0.065 | 0.032 |
| **Mean** | **0.193** | **0.258** | **0.187** | **0.184** | **0.160** | **0.160** |

Table 6: Abnormality-wise F1 scores on the RAD-ChestCT dataset (external test set), along with the prevalence ratio of each abnormality. BTB3D demonstrates strong generalization performance, particularly with the $16\times16\times8$ variant, achieving the highest F1 score across most categories.

| Abnormality | Ratio | Ours-16 | Ours-8 | CT-CHAT | Merlin | CT2Rep |
|---|---|---|---|---|---|---|
| Medical material | 0.327 | 0.235 | 0.154 | 0.000 | 0.107 | 0.000 |
| Calcification | 0.706 | 0.671 | 0.406 | 0.567 | 0.359 | 0.434 |
| Cardiomegaly | 0.109 | 0.187 | 0.151 | 0.181 | 0.031 | 0.010 |
| Pericardial effusion | 0.155 | 0.193 | 0.101 | 0.007 | 0.047 | 0.089 |
| Hiatal hernia | 0.117 | 0.149 | 0.087 | 0.149 | 0.136 | 0.173 |
| Lymphadenopathy | 0.165 | 0.252 | 0.227 | 0.121 | 0.176 | 0.000 |
| Emphysema | 0.273 | 0.201 | 0.172 | 0.412 | 0.242 | 0.142 |
| Atelectasis | 0.298 | 0.350 | 0.265 | 0.387 | 0.194 | 0.153 |
| Lung nodule | 0.802 | 0.424 | 0.408 | 0.721 | 0.495 | 0.068 |
| Lung opacity | 0.539 | 0.542 | 0.389 | 0.140 | 0.394 | 0.560 |
| Pulmonary fibrotic sequela | 0.132 | 0.204 | 0.139 | 0.026 | 0.174 | 0.157 |
| Pleural effusion | 0.200 | 0.290 | 0.199 | 0.043 | 0.135 | 0.032 |
| Peribronchial thickening | 0.085 | 0.117 | 0.064 | 0.000 | 0.095 | 0.039 |
| Consolidation | 0.139 | 0.217 | 0.145 | 0.108 | 0.149 | 0.264 |
| Bronchiectasis | 0.158 | 0.128 | 0.084 | 0.034 | 0.122 | 0.000 |
| Interlobular septal thick. | 0.069 | 0.091 | 0.075 | 0.008 | 0.057 | 0.000 |
| **Mean** | **0.267** | **0.266** | **0.192** | **0.182** | **0.182** | **0.133** |

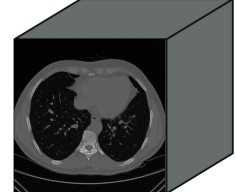

**Ground Truth**

**Findings:** Trachea and both main bronchi are open. No occlusive pathology was detected in the trachea and both main bronchi. There are linear atelectasis in the right lung middle lobe and medial upper lobe anterior segment, and left lung upper lobe lingular segment. Minimal emphysematous changes were observed in both lungs. There is a millimetric nodule in the upper lobe of the right lung. No mass or infiltrative lesion was detected in both lungs. As far as can be observed: Heart contour and size are normal. No pleural or pericardial effusion was detected. The widths of the mediastinal main vascular structures are normal. No pathologically enlarged lymph nodes were detected in the mediastinum and hilar regions. No pathological wall thickness increase was observed in the esophagus within the sections. No upper abdominal free fluid-collection was detected in the sections. No enlarged lymph nodes in pathological dimensions were detected. In the upper abdominal organs within the sections, there is no mass with distinguishable borders as far as it can be observed within the borders of non-enhanced CT. Thoracic vertebral corpus heights, alignments and densities are normal. There are osteophytes in the vertebral corpus corners.
**Impression:** Minimal emphysematous changes in both lungs. Linear atelectasis in both lungs. Millimetric nodule in the upper lobe of the right lung.

**Ours (16x16x8)**

**Findings:** Trachea, both main bronchi are open. Mediastinal main vascular structures, heart contour, size are normal. Thoracic aorta diameter is normal. Pericardial effusion-thickening was not observed. Thoracic esophageal calibration was normal and no significant tumoral wall thickening was detected. No enlarged lymph nodes in prevascular, pre-paratracheal, subcarinal or bilateral hilar-axillary pathological dimensions were detected. When examined in the lung parenchyma window; Minimal paraseptal emphysematous changes are observed in the right lung lower lobe superior segment posterolateral. Linear atelectasis changes are observed in the right lung middle and upper lobe anterior segment. There is a nodule with a diameter of 3 mm in the upper lobe of the right lung. There was no finding compatible with pleural effusion, pneumothorax or pneumonia in both lungs. Upper abdominal organs included in the sections are normal. No space-occupying lesion was detected in the liver that entered the cross-sectional area. Bilateral adrenal glands were normal and no space-occupying lesion was detected. Mild degenerative changes are observed in the bone structure entering the examination area. Vertebral corpus heights are preserved.
**Impression:** Minimal paraseptal emphysematous changes in the right lung lower lobe superior segment posterolateral, linear subpleural atelectasis changes in the right lung middle and upper lobe anterior segment, a nonspecific nodule in the upper lobe of the right lung; no findings in favor of pneumonia were detected.

**Ours (8x8x8)**

**Findings:** Trachea and both main bronchi are open. No occlusive pathology was detected in the trachea and both main bronchi. The left lung is in the collumnar fashion and there are calcific nodules in the interlobular fissure on the left. The sequelae were evaluated in favor of changes. There are linear atelectasis in both lungs. Emphysematous changes were observed in both lungs. No mass or infiltrative lesion was detected in both lungs. Mediastinal structures cannot be evaluated optimally because contrast material is not given. As far as can be observed: Heart contour and size are normal. There is minimal pericardial effusion. The widths of the mediastinal main vascular structures are normal. There are no pathologically enlarged lymph nodes in the mediastinum and hilar regions. There is no pathological wall thickness increase in the esophagus within the sections. No upper abdominal free fluid-collection was detected in the sections. No enlarged lymph nodes in pathological dimensions were detected. No fractures or lytic-destructive lesions were observed in the bone structures within the sections.
**Impression:** Emphysematous changes in both lungs. Locally linear atelectasis in both lungs. Minimal pericardial effusion.

**CT-CHAT**

**Findings:** Trachea and both main bronchi are open. No occlusive pathology was detected in the trachea and both main bronchi. There are minimal emphysematous changes in both lungs. There are millimetric nonspecific nodules in both lungs. No mass or infiltrative lesion was detected in both lungs. Mediastinal structures cannot be evaluated optimally because contrast material is not given. As far as can be observed: Heart contour and size are normal. No pleural or pericardial effusion was detected. The widths of the mediastinal main vascular structures are normal. No pathologically enlarged lymph nodes were detected in the mediastinum and hilar regions. No pathological wall thickness increase was observed in the esophagus within the sections. No upper abdominal free fluid-collection was detected in the sections. No enlarged lymph nodes in pathological dimensions were detected. In the upper abdominal organs within the sections, there is no mass with distinguishable borders as far as it can be observed within the borders of non-enhanced CT. Thoracic vertebral corpus heights, alignments and densities are normal. Intervertebral disc distances are preserved. The neural foramina are open. No lytic-destructive lesions were detected in the bone structures within the sections.
**Impression:** Minimal emphysematous changes in both lungs. Millimetric nodules in both lungs.

**Merlin**

**Findings:** Trachea, both main bronchi are open. Mediastinal main vascular structures, heart contour, size are normal. Thoracic aorta diameter is normal. Pericardial effusion-thickening was not observed. Thoracic esophagus calibration was normal and no significant tumoral wall thickening was detected. No enlarged lymph nodes in prevascular, pre-paratracheal, subcarinal or bilateral hilar-axillary pathological dimensions were detected. When examined in the lung parenchyma window; Aeration of both lung parenchyma is normal and no nodular or infiltrative lesion is detected in the lung parenchyma. Pleural effusion-thickening was not detected. Upper abdominal organs included in the sections are normal. No space-occupying lesion was detected in the liver that entered the cross-sectional area. Bilateral adrenal glands were normal and no space-occupying lesion was detected. Bone structures in the study area are natural. Vertebral corpus heights are preserved.
**Impression:** Thoracic CT examination within normal limits.

**CT2Rep**

**Findings:** Trachea and both main bronchi are open. No occlusive pathology was detected in the trachea and both main bronchi. There is minimal bronchiectasis in the central parts of both lungs. Occasionally, linear atelectasis was observed in both lungs. There are emphysematous changes in both lungs. In the right lung, there are millimetric nodules with ground-glass areas around some of them. When evaluated together with the patient's primary disease, these appearances were primarily evaluated in favor of metastases. It is recommended that the patient be evaluated together with previous examinations, if any. There is no mass or infiltrative lesion in both lungs. Mediastinal structures cannot be evaluated optimally because contrast material is not given. As far as can be observed, the heart is larger than normal. There is no pleural or pericardial effusion. There are millimetric atheroma plaques in the aorta. No pathologically enlarged lymph nodes were detected in the mediastinum and hilar regions. There is no pathological wall thickness increase in the esophagus within the sections. No upper abdominal free fluid collection was detected in the sections. No enlarged lymph nodes in pathological dimensions were detected. There are sometimes millimetric hypodense lesions in the bone structures within the sections. Although the described appearances cannot be characterized because they are very small, it was thought that the presence of primary disease could indicate metastases of these appearances. Further investigation is recommended.

Figure 5: Example of radiology report generation for the same 3D chest CT scan using our BTB3D method (both $16\times16\times8$ and $8\times8\times8$ variants) compared to baseline models (CT-CHAT, Merlin, CT2Rep) and the ground truth report. Key phrases from the ground truth are highlighted and matched across model outputs using consistent colors to indicate alignment. Our BTB3D framework, especially the higher compression rate variant, produces more detailed, clinically relevant, and accurate radiology reports, showing superior coverage of anatomical structures and abnormalities.

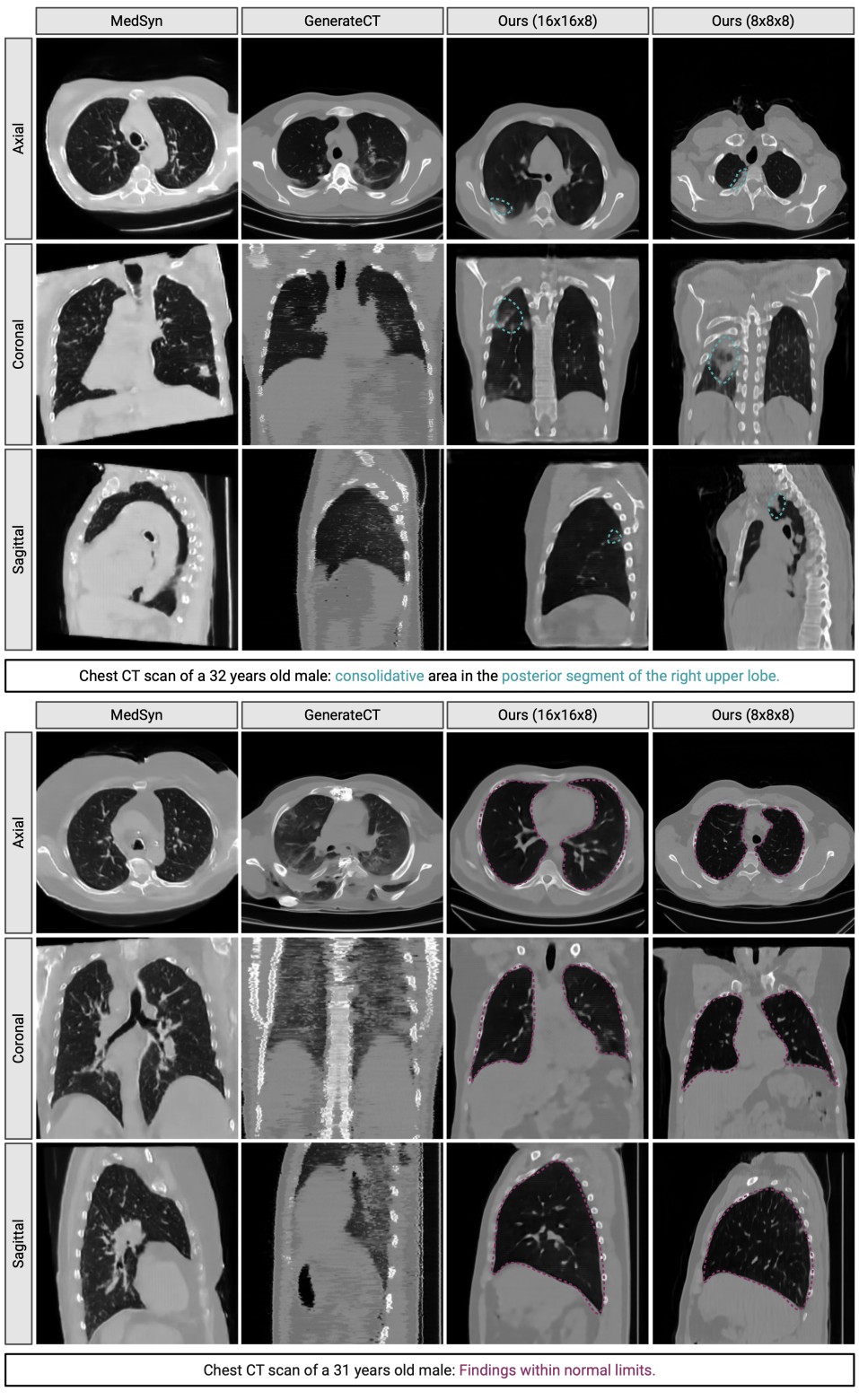

Figure 6: Text-conditioned CT generation results for two clinical prompts using MedSyn, GenerateCT, and our BTB3D models. For each case, we show one representative slice per anatomical plane. The lower-compression variant of BTB3D produces the most consistent volumes, demonstrating superior alignment with the prompt. Prompts and corresponding anatomical regions are highlighted using color overlays. Ground-truth volumes are omitted, following standard practice in generative tasks.

