# OpenReview forum: "Better Tokens for Better 3D: Advancing Vision-Language Modeling in 3D Medical Imaging"
_NeurIPS.cc/2025/Conference — NeurIPS 2025 poster_

### Official Review · Reviewer_kRtj · 2025-06-12

[review text omitted: it was posted to a different submission]

---

> ### Author Rebuttal · Authors · 2025-07-30
>
> ## Reviewer kRtj
>
> We sincerely thank the reviewer for their time. However, **this review appears to have been mistakenly assigned to our submission and does not discuss our work**.
>
> Our paper, **"Better Tokens for Better 3D (BTB3D): Advancing Vision-Language Modeling in 3D Medical Imaging"**, focuses specifically on:
> - Improving 3D vision-language modeling for **chest CT** scans,
> - Introducing a **causal convolutional tokenizer** and a **three-stage curriculum** for scalable long-sequence processing,
> - Enabling key downstream tasks such as **radiology report generation** and **text-to-CT synthesis** in 3D,
> - Addressing the unique challenges of **3D medical imaging**.
>
> In contrast, the review discusses a different paper, named **MedCoT**, focused on:
> - A **text and image compression framework** for multimodal medical tasks,
> - Experiments conducted primarily on **2D X-ray datasets** such as MIMIC-CXR and PathVQA,
> - Use of **medical ontology-based text compression** and **abstract visual prompting**,
> - Tasks including **question-answering, localization, and multi-task learning**.
>
> We are confident that this is a mismatched review, likely intended for a different submission in the reviewer’s stack. We understand that such mismatches can occur in large-scale reviewing pipelines. We kindly ask the program chairs and the reviewer to reassess and either reassign or disregard this review, as appropriate.

---

### Official Review · Reviewer_b6g3 · 2025-07-02

**Clarity:** 3
**Significance:** 3
**Originality:** 3
**Rating:** 5
**Confidence:** 4

**Summary:**

This paper introduces BTB3D (Better Tokens for Better 3D), a encoder-decoder framework for vision-language modeling in 3D medical imaging. The causal convolutional architecture processes CT scans slice-by-slice while maintaining temporal consistency, combined with wavelet-based compression and lookup-free quantization. The authors propose a three-stage training curriculum that progressively scales from short 9-slice subvolumes to full 300+ slice CT scans, addressing the computational challenges of training on high-resolution 3D medical data. The framework demonstrates good performance on two key tasks: radiology report generation (and text-conditioned CT synthesis.

**Questions:**

- Have you conducted any qualitative evaluation with radiologists? what features does the model fail to reconstruct properly etc?
- Whats the performance of other similar tokenziers e.g cosmos's CV/DV tokenizers on this task?

**Ethical Concerns:**

["NO or VERY MINOR ethics concerns only"]

**Final Justification:**

I'm inclined to improve the score of the paper to 5, giving the additional human eval, and robust rebuttal to the concerns raised.

**Limitations:**

- clear dataset bias, nonetheless, this is understandable given the requirements for the data and the nature of medical datasets.
- no clear clinical evaluation, beyond downstream use and upstream reconstruction metrics.

**Paper Formatting Concerns:**

- Figure quality: Figure 2's reconstruction comparisons are difficult to distinguish at the current resolution. Consider using zoomed insets to highlight differences. Also it would seem for the specific example chosen stage 2's result looked better so this may be crucial to see the failures of other stages
Missing details: The discriminator architecture for adversarial training is mentioned but never specified in the main text

**Quality:**

3

**Strengths And Weaknesses:**

Strengths:
- The paper clearly articulates why contrastive learning fails for 3D medical VLMs (semantic alignment issues, memory constraints) and proposes a reconstruction-based alternative that addresses these limitations systematically.
- They develop a good 2D/3D causal convolution based tokenizer, with good temporal consistency similar to Cosmos tokenizers
- Novel three-stage curriculum effectively bridges the gap between computational constraints during training and the need to process full-length volumes at inference.
- The paper evaluates on multiple downstream tasks with strong results.

Weaknesses:
- While the three-stage training is evaluated, other design choices lack thorough ablation.
-  Minor: Despite addressing efficiency, training still requires 64 H100 GPUs. This limits accessibility.
- No clinical validation with experts.
- No comaprisim with non-medical tokenizers

---

> ### Author Rebuttal · Authors · 2025-07-30
>
> ## Reviewer b6g3
>
> We sincerely thank the reviewer for the thoughtful and constructive review. We especially appreciate your emphasis on the key strengths of our paper:
> - The detailed critique of contrastive learning limitations in 3D medical vision-language models;
> - The development of a causal convolution-based tokenizer with strong temporal consistency;
> - The introduction of our novel three-stage training curriculum for full-volume CT processing;
> - The comprehensive evaluations across both report generation and text-to-CT synthesis tasks.
>
> We are also grateful for your understanding of our dataset limitations, which stem from the current scarcity of publicly available large-scale paired datasets in medical imaging.
>
> Below we provide a point-by-point response to your concerns and suggestions:
>
> ---
>
> ### **1. Other Design Choices Lack Thorough Ablation**
>
> We acknowledge that additional ablations for architectural components would strengthen our findings. In this work, we prioritized a **comprehensive ablation of our main contribution: the three-stage training strategy**, as shown in **Table 1 and Figure 2** of the main paper.
>
> Due to the **high computational cost** (around 1 week with 64× H100 GPUs), we were unable to thoroughly ablate every individual design choice. Our goal was to achieve strong performance in demanding multimodal tasks such as **report generation** and **text-to-CT synthesis**, which require long-range spatial-temporal modeling and high semantic fidelity.
>
> Nonetheless, we recognize the importance of additional ablations and appreciate the reviewer’s suggestion.
>
> ---
>
> ### **2. High Training Cost Limits Accessibility**
>
> You are absolutely right that BTB3D is computationally intensive during training. Our current setup requires approximately **one week of training on 64× H100 GPUs**.
>
> However, we designed BTB3D to optimize for **clinical-scale downstream performance**, not training efficiency. Thanks to our **three-stage curriculum and overlapping tiling**, **inference is significantly more affordable**:
> - **Radiology report generation** for a full 3D CT volume takes **<18 seconds** on a single H100 GPU;
> - **Text-to-CT generation** takes **<12 seconds** on a single H100 GPU.
>
> These results are highly valuable in the 3D medical imaging field. For context, one of the baselines, **GenerateCT**, reports an inference time of **184 seconds** per scan on an A100 80GB GPU, compared to BTB3D’s **<12 seconds** on a single H100.
>
> While clinical deployment still requires further steps, these results demonstrate that **BTB3D is a promising direction for scalable, low-latency 3D medical VLMs**.
>
> ---
>
> ### **3. No Clinical Validation with Experts**
>
> We strongly agree that **metrics alone are insufficient** to assess clinical usability or realism in report generation and CT synthesis.
>
> As part of this revision, we conducted **clinical expert evaluations** on both tasks:
>
> #### **Radiology Report Generation Evaluation:**
>
> We selected a **subset of 150 CT scans** from the CT-RATE test set for expert evaluation. Since the **full test set contains over 3,000 scans**, evaluating all was infeasible.
>
> To ensure fairness and diversity (rather than random sampling), we adopted a **stratified sampling strategy based on LLaMA Score**, using an LLM-based automated evaluation metric introduced in the CT-RATE benchmark [3]. This metric scores each generated report from 0–10 based on clinical plausibility, using a pretrained LLM as an evaluator. We calculated LLaMA scores across the entire CT-RATE test set using our BTB3D 16×16×8 variant.
>
> We then sampled 50 reports from each LLaMA score tier: **Low** (0–3), **Medium** (4–6), **High** (7–10)
>
> These 150 reports were assessed by a **board-certified radiologist**, who scored each on a scale of 0–10 based on **diagnostic relevance**, **semantic accuracy**, and **clinical coherence**.
>
> ---
>
> |                | CT2Rep | Merlin | CT-CHAT | BTB3D |
> |----------------|--------|--------|---------|--------|
> | **Expert Score**| 2.74   | 3.82   | 4.52    | **5.46** |
>
> ---
>
> These results further show that **BTB3D sets a new state-of-the-art for radiology report generation in 3D CT**. However, our radiologist emphasized that **even the best models remain far from clinical deployment**.
>
> Specifically, the expert noted that current models can **identify gross abnormalities** but **struggle with nuanced details**, such as:
> - Accurate **counts** of findings (e.g., number of nodules),
> - Precise **measurements** (e.g., sizes),
> - Consistent description of **pathological features** (e.g., shape, margins, density).
>
> This confirms that while BTB3D leads among existing approaches, **substantial progress is still needed for reliable clinical use**.
>
> ---
>
> #### **Text-to-CT Generation Evaluation:**
>
> Using the same 150 prompts, we generated chest CT scans using **BTB3D, GenerateCT, and MedSyn**. The same radiologist was asked to evaluate:
> - The **visual realism** of each 3D volume,
> - The **alignment** between the generated scan and the conditioning text.
>
> |          | GenerateCT | MedSyn | BTB3D |
> |---------------|------------|--------|--------|
> | **Realism**       | 2.90       | 2.47   | **6.58** |
> | **Alignment**     | 3.74       | 3.66   | **4.12** |
>
> BTB3D achieved the highest scores in both realism and alignment, further validating its ability to model high-fidelity semantic structure from text. However, as noted by the expert, text-to-CT alignment remains an area for improvement, as the generated volumes were not always well-aligned with the conditioning prompts.
>
> *We used the 16×16×8 variant of BTB3D for report generation and 8×8×8 variant for text-to-CT synthesis.*
>
> ---
>
> ### **4. No Comparison with Non-Medical Tokenizers (e.g., Cosmos)**
>
> Thank you for highlighting this important point.
>
> We also explored using **Cosmos CV/DV tokenizers** (which are state-of-the-art in video modeling). However, we encountered several barriers:
>
> - **Domain mismatch**: Cosmos models are pretrained on **RGB video**, whereas CT scans are **grayscale medical volumes**.
> - **Sequence length limitations**: Cosmos is designed for sequences up to **121 frames**, while CT-RATE scans can exceed **600 slices**.
> - **Training code unavailability**: Public repositories only support **inference**, making it infeasible to fine-tune Cosmos on CT datasets.
>
> In our initial evaluations, Cosmos models performed **worse** on:
> - **Reconstruction quality**,
> - **Report generation accuracy**,
> - **Text-conditioned CT generation fidelity**.
>
> We attribute this gap primarily to **training domain mismatch**, not architecture flaws. As **fine-tuning was not feasible**, we refrained from including Cosmos in the main paper to avoid unfair comparisons.
>
> That said, we agree with the reviewer and documenting these findings in the supplementary will provide valuable context. We will add a section comparing Cosmos and BTB3D in the revised appendix.
>
> ---
>
> ### **5. Formatting and Presentation Issues**
>
> We appreciate your helpful suggestions to improve readability and clarity:
>
> - **Figure 2**: We will revise the figure by adding **zoomed-in insets** and adjusting **contrast levels** to better highlight reconstruction differences between stages. We will also clarify cases where Stage 2 outperforms Stage 3 on specific slices.
> - **Discriminator architecture**: We appreciate this comment, the adversarial discriminator was very briefly mentioned in Section 4.1 but lacked full details. We will update the **Methods section** to include complete architectural specifications (layers, kernel sizes, etc.) for reproducibility.
>
> ---
>
> Once again, we thank the reviewer for your fair and insightful feedback. Your comments helped us strengthen both the **experimental evaluation** and **clinical relevance** of BTB3D, and we hope the additional clarifications address your concerns.

---

> > ### Comment · Reviewer_b6g3 · 2025-08-04
> >
> > I appreciate the rebuttal, esp. the expert eval. I still believe in my assessment of the level of contributions and will keep the score unchanged.

---

> ### Author Response · Authors · 2025-08-06
>
> We sincerely thank you for your positive score and constructive feedback, which helped us strengthen the paper. We especially appreciate your acknowledgment of our rebuttal and expert evaluations, as well as the time you took to review our clarifications.
>
> Our core contribution is a scalable tokenization strategy for 3D medical imaging, built on designing a causal convolutional encoder-decoder and a three-stage training scheme with overlapping temporal tiling. This enabled strong improvements across both key tasks:
>
> - **Radiology report generation**: BTB3D achieved a **40% relative F1 gain** over CT-CHAT, with improved anatomical grounding and semantic accuracy.
> - **Text-to-CT generation**: BTB3D reduced **FID by 75%** and **FVD by over 50%** compared to GenerateCT and MedSyn, with superior expert-rated realism and alignment.
>
> These results underscore the importance of scalable tokenizers (beyond contrastive learning or large LLMs) for long, high-resolution 3D scans.
>
> If any concerns remain, we’d be grateful for further feedback to continue improving the work. Thank you again for your insight and support.

---

### Official Review · Reviewer_PJbc · 2025-07-07

**Clarity:** 3
**Significance:** 3
**Originality:** 2
**Rating:** 4
**Confidence:** 4

**Summary:**

The authors propose BTB3D, a 3d convolutional encoder-decoder that converts CT scans into a sequence of discrete and frequency aware tokens. There are primarily 4 components to their methodology - a Haar wavelet processor that shrinks the volume while accumulating channels with frequency information, a causal 3d CNN, a lookup free binary quantization that improves speed and memory efficiency, and a three-stage training setup. The training objectives basically (1) enable local slice reconstruction through short-volume pretraining, (2) overlapping temporal tiling that enforces consistency across slices of the volume, and (3) decoder-only fine tuning on the full scans. The resulting tokens are evaluated on the task of report generation and text-to-CT synthesis. The results show that BTB3D beats the baselines such as CT-CHAT, Merlin and Generate-CT across various metrics, highlighting the strengths of the proposed method

**Questions:**

1. Is there a quick way to show/discuss generalization beyond chest CT. What changes are required for a different anatomy or modality - slice window, increasing channel dimension etc. Showing transfer to other body regions or modality would raise the significance of the method
2. Does changing the tiling length affect downstream metrics? I wonder if variable slice windows based on which slice range the anatomy is usually located in could affect reconstruction
3. A few examples of generated reports vs. those from the baselines such as merlin or CT-Chat would be useful to see, especially since the clinical accuracy metrics (P/R) of the baselines are better than those from BTB3D

**Ethical Concerns:**

["NO or VERY MINOR ethics concerns only"]

**Final Justification:**

The authors provide detailed clarification to the questions I had. Particularly, the experiments on MRI data established cross -modality performance and shows promise without fine-tuning. I've read the other reviews and responses and doing another detailed pass over the paper including supplementary Figure 2., I am happy to increase my rating to 4.

**Limitations:**

Yes, the authors note the limitations in the checklist

**Paper Formatting Concerns:**

No major formatting issues. I haven't checked for any grammatical errors, but the paper reads well across sections, with figures and tables clearly formatted

**Quality:**

3

**Strengths And Weaknesses:**

Strengths:
1. The writing is generally clear, concise, and the figures are informative
2. Each component of the encoder-decoder model is clearly explained. The causal 3d CNN design enables a single model to operate on 2d slices or full 3d volumes. The novelty of this work is in the three stage training pipeline, which solves learning over longer CT sequences through overlapping tiling and decoder only fine tuning. As Table 1 shows, stage 2 and 3 enable dramatic improvement in reconstruction metrics.
3. The same discrete tokens derived from the pretrained model are used by an (1) LLM for report generation, (2) a transformer decoder with cross attention, for CT synthesis. This demonstrates that the tokenizer is versatile and agnostic to the downstream head.
4. The report generation performance on CT-RATE across NLG metrics is higher than that of the baselines. Similarly, the CT generation metrics of BTB3D is better than generateCT and MedSyn across the board.
5. On the implementation side, dataset and training details are clearly provided along with resource requirements and tradeoffs between various components in the model in terms of speed and memory

Weaknesses:
1. The novelty lies in the three stage training pipeline, while the encoder decoder components themselves are an application of prior work (related work is mentioned in the paper under each subsection in methods) to the CT dataset. The title of the paper “BTB3D: Advancing VLM in 3D Medical Imaging” suggests the applicability of the method across other modalities in medical imaging, which is an overstatement. There is no experiment to show the usage of the method on MRI data, for example. The usage of a fixed 9-slice window, and the Haar wavelet transform seem to be well suited and chosen for chest CT data - there is no evidence this holds for other modalities and no experiments are provided on how to tune it to other modalities. The title of the paper has to match the empirical scope shown in it.
2. The authors note the above limitation in the checklist and attribute it to the lack of scaled image-report paired datasets. However, none of the pretraining stages, which is the core contribution, require paired reports. Only the downstream tasks of report gen would be a blocker. Datasets such as BraTS and fastMRI are widely used in the medical imaging community.

---

> ### Author Rebuttal · Authors · 2025-07-30
>
> ## Reviewer PJbc
>
> We sincerely thank the reviewer for the thoughtful and constructive review. We especially appreciate your detailed emphasis on the **clarity of our writing**, the **novelty and effectiveness of the three-stage training**, and the **strong performance across both report generation and text-to-CT synthesis**. Your recognition of the **modular versatility of our token representations** and the **transparency of our implementation** is deeply encouraging.
>
> Below we provide point-by-point responses to your comments and suggestions:
>
> ---
>
> ### **1. Scope Overstatement in Title & Generalization Beyond Chest CT**
> > *“The title ... suggests the applicability ... across other modalities ... which is an overstatement ... there is no experiment to show ... on MRI data ... usage of a fixed 9-slice window and Haar transform seem CT-specific.”*
> > *“None of the pretraining stages require paired reports ... datasets such as BraTS and fastMRI are widely used.”*
> > *“Is there a quick way to show/discuss generalization beyond chest CT?”*
>
> We thank the reviewer for this valuable observation. You are absolutely right, our **experiments are currently limited to chest CT**, as stated in Limitations section (Appendix A.1) and throughout the main paper, due to the lack of large-scale, report-paired datasets in other modalities. While we have noted this limitation throughout the paper, we acknowledge that the title may overstate the generality of our approach. The phrase 3D medical imaging can be interpreted more broadly than our current empirical scope.
>
> While paired reports are not required for the pretraining stages, our **main motivation** was to enable **vision-language modeling (VLM)** in 3D medical imaging (specifically, **report generation** and **text-to-volume synthesis**) both of which require report-paired data. Unfortunately, datasets such as **BraTS [1]**, **fastMRI [2]**, and **FOMO-MRI [3]** do not provide such supervision and are therefore unsuitable for these tasks.
>
> That said, we agree with the reviewer that our **main contribution lies in the three-stage training strategy**, and that its effectiveness **can and should be tested across modalities**. To address this, we conducted additional experiments to evaluate the generalizability of our three-stage pretraining pipeline on:
> - **FOMO-MRI (T1-weighted brain MRIs, n=8857) [3]**
> - **TotalSegmentator full-body CT scans (n=1228) [4]**
>
> We used our pretrained BTB3D encoder-decoder **without any fine-tuning**, and evaluated reconstruction quality across stages using PSNR, SSIM, and MSE. Results are shown below:
>
> ### Brain MRI (FOMO-MRI, T1-only)
>
> | **Stage**  | **PSNR ↑** (16×16×8) | **SSIM ↑** (16×16×8) | **MSE ↓** (16×16×8) | **PSNR ↑** (8×8×8) | **SSIM ↑** (8×8×8) | **MSE ↓** (8×8×8) |
> |------------|----------------------|----------------------|---------------------|--------------------|--------------------|-------------------|
> | Stage 1    | 12.60                | 0.409                | 0.223               | 8.05               | 0.074              | 0.627             |
> | Stage 2    | 24.95                | 0.825                | 0.013               | 31.14              | 0.928              | 0.004            |
> | Stage 3    | **27.00**            | **0.889**            | **0.008**           | **31.27**          | **0.931**          | **0.003**        |
>
> ---
> ### Full-Body CT (TotalSegmentator)
>
> | **Stage**  | **PSNR ↑** (16×16×8) | **SSIM ↑** (16×16×8) | **MSE ↓** (16×16×8) | **PSNR ↑** (8×8×8) | **SSIM ↑** (8×8×8) | **MSE ↓** (8×8×8) |
> |------------|----------------------|----------------------|---------------------|--------------------|--------------------|-------------------|
> | Stage 1    |  11.43  | 0.197  | 0.297 | 8.86  | 0.039  | 0.529             |
> | Stage 2    | 15.48  | 0.371  | 0.126               | 18.19  | 0.460  | 0.075 |
> | Stage 3    |**17.17** | **0.429**  | **0.071**           | **19.51** | **0.498**  | **0.058** |
>
>
> ---
>
> These results confirm that our **progressive three-stage training strategy consistently improves performance**, especially through **Stage 2's overlapping tiling** (which is our main contribution). While these are not full VLM evaluations, they offer promising signals that BTB3D can generalize across modalities and body regions.
>
> We also acknowledge that further **adaptations may be necessary** for full extension to VLM tasks in other modalities. For example:
> - Multi-series MRI (e.g., T1/T2/FLAIR) may require an updated input channel strategy.
> - The Haar wavelet transform may not be optimal in densely structured anatomical domains such as brain or abdominal MRI. Ablation studies will be required to explore this further.
>
> We hope that these additional results offer meaningful insight and will help inform future applications of BTB3D to other modalities.
>
> ---
>
> ### **2. Impact of Tiling Length**
> > *“Does changing the tiling length affect downstream metrics?”*
>
> Thank you for raising this important question, we have also considered this in our internal discussions. In our current design, we tile with **overlapping windows sharing only one slice**, which provides a reasonable balance between **temporal consistency** and **computational cost**.
>
> Increasing the overlap (e.g., to 3 slices) could:
> - Enhance temporal continuity further;
> - Improve capture of localized structures (e.g., the diaphragm or cardiac apex).
>
> However, doing so would **significantly increase memory usage and training time**, which are already considerable with our 3-stage setup. Given that we already significantly outperform all baselines, we did not increase overlap in this version. Nevertheless, we believe your suggestion is highly valid and worthy of future investigation if needed, especially as computational resources become more accessible.
>
> ---
>
> ### **3. Examples of Generated Reports**
> > *“A few examples of generated reports vs. baselines ... would be useful to see.”*
>
> We completely agree with the reviewer. Generated reports are an important qualitative indicator. As such, we already included **side-by-side comparisons** in **Supplementary Figure 2**, showing:
> - The **ground-truth report**,
> - Reports generated by **BTB3D, CT-CHAT, CT2Rep, and Merlin**, with consistent color-coding by abnormality.
>
> We acknowledge that placing them in the appendix may have reduced their visibility, and we will consider bringing selected examples into the main paper in the revised version.
>
> ---
>
> **References:**
> [1] Menze et al., IEEE TMI 2015. *The Multimodal Brain Tumor Image Segmentation Benchmark (BraTS)*
> [2] Zbontar et al., arXiv 2018. *fastMRI: An Open Dataset and Benchmarks for Accelerated MRI*
> [3] Munk et al., arXiv 2025. *A Large‑Scale Heterogeneous 3D Magnetic Resonance Brain Imaging Dataset for Self‑Supervised Learning (FOMO‑MRI)*
> [4] Wasserthal et al., Radiology: Artificial Intelligence 2023. *TotalSegmentator: Robust Segmentation of 104 Anatomical Structures in CT Images*
>
> ---
>
> Once again, we thank the reviewer for their evaluation. Your comments prompted us to strengthen our **cross-modality validation** and clarify our implementation choices. We hope these revisions demonstrate BTB3D’s potential more clearly and make a strong case for its contributions to vision-language modeling in 3D medical imaging.

---

> > ### Comment · Reviewer_PJbc · 2025-08-08
> >
> > I thank the authors for providing clarification to the questions I had. Particularly, the experiments on MRI data established cross -modality performance and shows promise without fine-tuning. I've read the other reviews and responses and doing another detailed pass over the paper including supplementary Figure 2., I am happy to increase my rating,

---

> > > ### Author Response · Authors · 2025-08-08
> > >
> > > Thank you for your thoughtful feedback and for revisiting our paper. Your suggestions significantly improved our work, especially around cross-modality evaluation. We sincerely appreciate the updated rating and your support.

---

> ### Author Response · Authors · 2025-08-06
>
> Dear Reviewer PJbc,
>
> Thank you once again for your thoughtful feedback and insightful questions. In our rebuttal we have:
>
> 1. **Demonstrated generalization beyond chest CT** by applying our models from the three-stage training pipeline to full-body CT (TotalSegmentator) and T1-weighted brain MRI (FOMO-MRI), showing consistent gains across stages.
> 2. **Clarified architectural generality**, noting that our design is generic and includes no CT-specific or chest-region assumptions, making it suitable for other imaging modalities and anatomical regions; parameters such as window size and input channels can be tuned as needed.
> 3. **Explained the tiling choice** and outlined the trade-offs between overlap length, memory use, and performance.
> 4. **Provided qualitative report comparisons**, already available in Supplementary Fig. 2; we acknowledge these might have been missed and will move representative examples into the main text for greater visibility.
>
> We hope these additions address your concerns. If any points would benefit from further clarification or discussion, please let us know, we greatly value your input.
>
> Thank you for helping us strengthen the manuscript.
>
> Best,
> The Authors

---

### Official Review · Reviewer_Ks7t · 2025-07-12

**Clarity:** 3
**Significance:** 2
**Originality:** 2
**Rating:** 4
**Confidence:** 4

**Summary:**

This paper introduces BTB3D, a causal convolutional encoder-decoder framework for vision-language modeling in 3D medical imaging. To generate compact and semantically meaningful volumetric tokens, the model combines wavelet-based compression, causal 3D convolutions, and lookup-free quantization. To address limitations of contrastive pretraining misaligned with clinical semantics, BTB3D adopts a three-stage training strategy: (1) short-volume pretraining, (2) overlapping temporal tiling, and (3) long-context decoder refinement. Experimental results demonstrate that BTB3D achieves state-of-the-art performance on both radiology report generation and text-conditioned CT synthesis tasks.

**Questions:**

1. Could the authors discuss existing efforts in unifying 2D and 3D modeling in medical imaging, and how BTB3D differs from or builds upon them?

2. Could the authors compare BTB3D’s tokens against alternative tokenization methods, such as VQ-VAE, CLIP-based features, or anatomy-aware tokens, to better support the "better token" claim?

3. Could the authors provide ablation studies comparing report generation and text-to-CT synthesis performance between single-stage and three-stage training? This would more directly validate the proposed training strategy for the intended applications.

**Ethical Concerns:**

["NO or VERY MINOR ethics concerns only"]

**Final Justification:**

I recommend the paper for acceptance because the authors have adequately addressed my concerns regarding: (1) how BTB3D relates to prior efforts on unifying 2D and 3D modeling in medical imaging; (2) the empirical support for the claim of using “better tokens” through comparisons with alternative tokenization strategies; and (3) the effectiveness of the proposed three-stage training strategy, validated through additional ablation studies on downstream tasks like report generation. Their rebuttal was thoughtful and thorough, and I believe the revised version will make a meaningful contribution to the field.

**Limitations:**

The paper includes a limitations discussion, noting the chest CT-only scope, computational requirements, and lack of clinical user studies.

**Paper Formatting Concerns:**

No.

**Quality:**

3

**Strengths And Weaknesses:**

**Strengths**:

-	The model achieves strong performance in both report generation and image synthesis tasks, outperforming state-of-the-art baselines in F1, BLEU, and FID/FVD metrics.
-	The causal convolutional design and wavelet-based compression allow training and inference on long CT sequences under limited memory.
-	The authors provide detailed implementation and training setup and claim to release code and pretrained weights, which supports future use and benchmarking.

**Weaknesses**:

-	**Incomplete related work discussion**: While the paper emphasizes the novelty of unifying 2D and 3D training and inference while producing compact, frequency-aware volumetric tokens, it does not adequately discuss prior work in 2D–3D unification in the medical imaging domain. Relevant approaches, such as [1] [2], are not reviewed. In addition, the related work section lacks coverage of tokenization techniques like [3-5]. This weakens the contextual positioning of the contribution.

-	**Insufficient validation of the proposed training strategy**: The effectiveness of the proposed three-stage training is primarily validated via reconstruction metrics (e.g., PSNR, SSIM). However, the downstream tasks of interest (i.e., radiology report generation and text-to-CT synthesis) depend on semantic alignment rather than pixel-wise fidelity. It is unclear whether better reconstruction directly leads to better language generation or synthesis. The lack of direct comparison on downstream task performance across different training stages weakens the claim that the three-stage training benefits the end tasks.

-	**Lack of evidence for the “better token” claim**: While the tokenization is central to the paper’s claims, there is no direct comparison to other tokenization strategies (e.g., VQ-VAE, CLIP embeddings) to validate its advantage.

[1] Xie, Yutong, et al. "Unimiss: Universal medical self-supervised learning via breaking dimensionality barrier." European Conference on Computer Vision. Cham: Springer Nature Switzerland, 2022.

[2] He, Xiaoxuan, et al. "Unified Medical Image Pre-training in Language-Guided Common Semantic Space." European Conference on Computer Vision. Cham: Springer Nature Switzerland, 2024.

[3] Zhang, Jiahui, et al. "Regularized vector quantization for tokenized image synthesis." Proceedings of the IEEE/CVF Conference on Computer Vision and Pattern Recognition. 2023.

[4] Tseng, Albert, et al. "Qtip: Quantization with trellises and incoherence processing." Advances in Neural Information Processing Systems 37 (2024): 59597-59620.

[5] Yao, Ting, et al. "Wave-vit: Unifying wavelet and transformers for visual representation learning." European conference on computer vision. Cham: Springer Nature Switzerland, 2022.

---

> ### Author Rebuttal · Authors · 2025-07-29
>
> ## Reviewer Ks7t
>
> We sincerely thank the reviewer for the thoughtful and constructive feedback, as well as for recognizing the strengths of our work, including the strong performance across tasks, the novel causal and frequency-aware tokenization strategy, the three-stage training scheme, and our detailed implementation and open-sourcing commitment.
>
> Below we provide detailed point-by-point responses to the reviewer’s concerns.
>
> ---
>
> ### **1. Incomplete Related Work Discussion**
> > *"Could the authors discuss existing efforts in unifying 2D and 3D modeling in medical imaging, and how BTB3D differs from or builds upon them?"*
> > *"The related work section lacks coverage of tokenization techniques like [3–5]. This weakens the contextual positioning of the contribution."*
>
> Thank you for highlighting this important point. We agree that the related work section can be improved to better ground our contributions. In our original submission, we focused primarily on two task-specific areas that we aimed to improve:
> - **Radiology report generation from 3D medical images**, and
> - **Text-conditional 3D medical image generation**
>
> This focus was motivated by our goal to advance these downstream applications specifically in the context of **3D chest CT**, a domain that only recently became accessible at scale thanks to the release of large multimodal datasets such as **CT-RATE [6]** and **Merlin [7]**. While the studies [1,2] suggested by the reviewer indeed explore 2D–3D unification, they were proposed before the availability of such datasets and do not directly target downstream vision-language tasks in 3D medical imaging.
>
> Nonetheless, we fully agree that two additional sections would enrich the related work, particularly if placed before the discussion of downstream tasks (i.e., report generation and text-to-CT generation), to better contextualize our contributions:
> - **Tokenization strategies in computer vision**: We will discuss strategies such as regularized vector quantization [3], QTip [4], and Wave-ViT [5], and explain how BTB3D adapts these concepts to the 3D CT domain, which involves long volumetric sequences and significant computational constraints.
> - **Unified 2D–3D pretraining in medical imaging**: This section will include discussions of Unimiss [1] and Unified Medical Image Pretraining [2]. Compared to these, BTB3D not only supports 2D/3D unification, but also scales to long, high-resolution 3D volumes and demonstrates strong performance on downstream VLM tasks such as report generation and text-to-volume synthesis. BTB3D’s key contribution is not only unification, but also scalability to long sequences, which is not addressed by these prior works.
>
> We will incorporate these additions and citations in the revised paper to strengthen the contextual grounding. We thank the reviewer for this helpful suggestion.
>
> ---
>
> ### **2. Insufficient Validation of the Three-Stage Training Strategy**
> > *"Could the authors provide ablation studies comparing report generation and text-to-CT synthesis performance between single-stage and three-stage training?"*
>
> Thank you for this valuable suggestion. We initially validated the three-stage training strategy through **reconstruction metrics** (PSNR, SSIM, MSE) and qualitative reconstructions (see Figure 2 in the main paper), assuming that improvements in fidelity would translate into better downstream performance.
>
> However, we agree that this assumption should be tested explicitly, particularly since downstream tasks (such as **report generation**) rely on **semantic alignment** rather than pixel-wise similarity.
>
> To this end, we conducted an ablation study evaluating **report generation performance** at each training stage using the BTB3D 16×16×8 model. The results below were computed on our internal validation set using F1 score, CRG score, and BLEU mean:
>
> | **Stage**     | **F1 ↑** | **CRG ↑** | **BLEU ↑** |
> |---------------|----------|-----------|------------|
> | Stage 1       | 0.154    | 0.349     | 0.176      |
> | Stage 2 & 3   | **0.258** | **0.370** | **0.305**  |
>
> Importantly, since Stage 3 only refines the decoder and does **not affect the encoder**, it has no impact on report generation, which relies on encoder features. Stage 3 is specifically designed to improve reconstruction quality and text-to-CT generation fidelity. Therefore, report generation performance remains unchanged between Stage 2 and Stage 3.
>
> The majority of the gains in report generation come from our **temporal overlapping tiling strategy introduced in Stage 2**, which enforces long-range consistency and better captures volumetric semantic context, essential for generating clinically plausible reports.
>
> These results confirm that our progressive training scheme (especially the temporal tiling) significantly enhances VLM performance. We will include these results and their implications in detail in the final version of the paper. We thank the reviewer for this valuable suggestion.
>
> ---
>
> ### **3. Lack of Evidence for the “Better Token” Claim**
> > *"Could the authors compare BTB3D’s tokens against alternative tokenization methods, such as VQ-VAE, CLIP-based features, or anatomy-aware tokens?"*
>
> We appreciate this constructive suggestion. In fact, our baselines **already incorporate diverse tokenization strategies**, enabling indirect yet meaningful comparisons:
> - **CT2Rep** employs a CapPa-style encoder [8].
> - **Merlin** and **CT-CHAT** rely on CLIP-like contrastive embeddings.
> - **GenerateCT** and **MedSyn** use VQ-VAE/VQ-GAN tokenizers.
>
> Since all models are trained on the **same CT-RATE dataset**, the observed performance gains of BTB3D (especially in **Tables 2–4**) can largely be attributed to its **superior tokenization approach**. For example, **CT-CHAT** and **BTB3D** use the **same LLM backbone (LLaMA 3.1-8B)**, with the **only difference** being the tokenization strategy. BTB3D outperforms CT-CHAT significantly, which underscores the effectiveness of our causal, frequency-aware 3D tokenization and three-stage training scheme.
>
> Specifically:
> - In **report generation**, BTB3D shows clear improvements in F1 and BLEU scores over CT-CHAT and Merlin, suggesting more effective language-guided representations.
> - In **text-to-CT synthesis**, BTB3D significantly outperforms GenerateCT and MedSyn in **FID**, **FVD**, and **CLIP scores**, confirming that its tokenization enhances both realism and semantic alignment.
>
> We agree that these comparisons were not emphasized clearly enough in the initial submission. In the revised version, we will:
> - Clarify that **tokenization is the only varying component** in models like BTB3D and CT-CHAT.
> - Explicitly link BTB3D’s improvements to its **discrete, causal, volumetric tokenization and three-stage training scheme**.
> - Include **t-SNE visualizations** of the learned token spaces (omitted here due to rebuttal policy) comparing BTB3D with other models. Preliminary results show that BTB3D produces more coherent and semantically clustered token embeddings. We believe these visualizations are particularly important: if we claim that BTB3D generates better tokens for 3D computed tomography, it is essential to provide empirical evidence (such as improved structure and interpretability in the embedding space) to substantiate this claim.
>
> ---
>
> ### **Limitations**
>
> We sincerely thank the reviewer for acknowledging our limitations section, described in detail in Appendix A.1, including the current restriction to chest CT, the absence of clinical user studies, and the computational demands of training.
>
> ---
>
> **References:**
> [1] Xie et al., ECCV 2022. *Unimiss: Universal medical self-supervised learning via breaking dimensionality barrier*
> [2] He et al., ECCV 2024. *Unified Medical Image Pre-training in Language-Guided Common Semantic Space*
> [3] Zhang et al., CVPR 2023. *Regularized vector quantization for tokenized image synthesis*
> [4] Tseng et al., NeurIPS 2024. *QTip: Quantization with trellises and incoherence processing*
> [5] Yao et al., ECCV 2022. *Wave-ViT: Unifying wavelet and transformers for visual representation learning*
> [6] Hamamci et al., arXiv 2024. *Developing Generalist Foundation Models from a Multimodal Dataset for 3D Computed Tomography*
> [7] Blankemeier et al., arXiv 2024. *Merlin: A Vision Language Foundation Model for 3D Computed Tomography*
> [8] Tschannen et al., NeurIPS 2024. *Image Captioners Are Scalable Vision Learners Too*
>
> ---
>
> We again thank the reviewer for the detailed and insightful feedback. We believe these suggestions will substantially improve the clarity and positioning of our work.

---

> ### Comment · Reviewer_Ks7t · 2025-08-05
>
> Thank you for your thorough responses, which address all of my concerns. I am confident that the revised version, as outlined in the rebuttal, will contribute to the advancement of 3D medical imaging research.

---

> ### Author Response · Authors · 2025-08-06
>
> Thank you for your thoughtful feedback and kind acknowledgment of our contribution to 3D medical imaging research. We are delighted that all concerns have been resolved and greatly appreciate the score update. Your comments have been invaluable in improving the clarity, depth, and overall quality of our manuscript. We sincerely appreciate your support.

---

### Decision · Program_Chairs · 2025-09-17

**Decision:**

Accept (poster)

**Comment:**

The paper proposes a causal convolutional encoder-decoder method that combines 2D and 3D training with inference for 3D vision language modelling. Furthermore, a three stage curriculum learning strategy is introduced to overcome the limitations of contrastive learning misalignment with clinical semantics.

Some notable strengths of the submission include:

- the model obtains strong performance on report generation and image synthesis tasks
- the paper is well-written and in the methodology, the core components are explained in detail and clarity
- the motivation for contrastive learning struggles for 3D medical VLMs
- a novel three stage curriculum learning strategy
- propose a adequate 2D/3D causal convolution based tokenizer

Among the concerns highlighted by reviewers, the important ones are:

- the related work is missing citations of some relevant works
- the validation of 3-stage curriculum learning strategy needs more analyses
- novelty mainly lies in 3-stage learning strategy while the rest of the design resembles with prior work
- some design choices are not well ablated
- there is mention of clinical validation with experts

After the author-reviewer discussion period, majority of the reviewers expressed their satisfaction with the responses of authors on the concerns. For example, the concerns of better positioning in the literature for 2D and 3D medical imaging, empirical validation of 3D tokens, the experiments on MRI data, improved validation of 3-state learning strategy, and clinical experts validation has been resolved. To this end, the AC decides to recommend acceptance of the paper. Authors are recommended to include important reviewers comments in the final version.